# Ebf3⁺ niche-derived CXCL12 is required for the localization and maintenance of hematopoietic stem cells

Taichi Nakatani[1,2,3], Tatsuki Sugiyama [1,2,3], Yoshiki Omatsu [1,2,3], Hitomi Watanabe[4], Gen Kondoh[4] & Takashi Nagasawa [1,2,3] ✉

Lympho-hematopoiesis is regulated by cytokines; however, it remains unclear how cytokines regulate hematopoietic stem cells (HSCs) to induce production of lymphoid progenitors. Here, we show that in mice whose CXC chemokine ligand 12 (CXCL12) is deleted from half HSC niche cells, termed CXC chemokine ligand 12 (CXCL12)-abundant reticular (CAR) cells, HSCs migrate from CXCL12-deficient niches to CXCL12-intact niches. In mice whose CXCL12 is deleted from all Ebf3⁺/leptin receptor (LepR)⁺ CAR cells, HSCs are markedly reduced and their ability to generate B cell progenitors is reduced compared with that to generate myeloid progenitors even when transplanted into wild-type mice. Additionally, CXCL12 enables the maintenance of B lineage repopulating ability of HSCs in vitro. These results demonstrate that CAR cell-derived CXCL12 attracts HSCs to CAR cells within bone marrow and plays a critical role in the maintenance of HSCs, especially lymphoid-biased or balanced HSCs. This study suggests an additional mechanism by which cytokines act on HSCs to produce B cells.

Most blood cells, including immune cells, are generated from hematopoietic stem cells (HSCs), which are maintained throughout life in the stromal microenvironment termed niches in the bone marrow[1–4]. HSC niches comprise support cells (niche cells) that produce cytokines essential for survival, proliferation, and/or differentiation of HSCs and their progeny. Initially, dispersed fibroblastic cells, termed CXC chemokine ligand 12 (CXCL12)-abundant reticular (CAR) cells and Nestin-GFP^high periarteriolar mesenchymal stem cells in the bone marrow cavity have been reported to be important for HSC niches[5,6]. Among various types of candidate niche cells, several lines of evidence have demonstrated that CAR cells, which overlap strongly with leptin receptor-expressing (LepR⁺) cells, are the population of mesenchymal stem cells and the major cellular component of niches essential for the HSC maintenance[5,7,8]. CAR cells are defined as a lineage that expresses much levels of platelet-derived growth factor receptor β (PDGFRβ),

HSC niche factors, including CXCL12 and SCF, and transcription factors, including Foxc1 and Ebf3 than any other types of cells[5,7–10]. However, it is not completely understood how HSC niche cells regulate HSCs to produce a specific lineage of immune cells, including B cells.

Cytokines are known to act on the lineage-committed progenitors to regulate their survival, proliferation, and/or differentiation. For example, interleukin-7 (IL-7), which is produced by about 62% of the CAR/LepR⁺ cells, is essential for the proliferation and differentiation of B cell progenitors, including pro-B and pre-B cells in the bone marrow[11]. In addition, it has been speculated that the cytokine produced by HSC niche cells may act on HSCs and/or multipotent hematopoietic progenitors to induce their differentiation into the lineage-committed progenitors.

The chemokine CXCL12 and its primary receptor CXCR4 are essential for the maintenance of HSCs and the production of B cells

[1]Laboratory of Stem Cell Biology and Developmental Immunology, Graduate School of Frontier Biosciences, Osaka University, Suita, Osaka 565-0871, Japan. [2]Laboratory of Stem Cell Biology, Graduate School of Medicine, Osaka University, Suita, Osaka 565-0871, Japan. [3]Laboratory of Stem Cell Biology and Developmental Immunology, WPI Immunology Frontier Research Center, Osaka University, Suita, Osaka 565-0871, Japan. [4]Center for Animal Experiments, Institute for Life and Medical Sciences, Kyoto University, Kyoto 606-8507, Japan. ✉e-mail: nagasawa.takashi.fbs@osaka-u.ac.jp

and their progenitors as well as plasmacytoid dendritic cells (pDCs) and NK cells[5,12–15] and CAR/LepR[+] cells are the major producer of CXCL12 within bone marrow[7]. However, it remains unclear how CXCL12 regulates both HSC maintenance and B cell production. It was reported that the number of HSCs was unaltered in bone marrow but increased in peripheral blood when CXCL12 was conditionally deleted from LepR[+] cells using LepR-Cre;CXCL12[f/-] mice. This suggests that LepR[+] cell-derived CXCL12 is essential to inhibit egress of HSCs from bone marrow but not to maintain HSCs[16,17]. However, since CXCL12 was deleted from about 70% of the CAR cells in the mutants probably due to lower LepR expression and recombination efficiency of LepR-Cre in early postnatal bone marrow[16,18], there is a possibility that CXCL12 production from a small subset of CXCL12-intact CAR cells contributes to the HSC maintenance.

In this study, we examine the role of CAR cell-derived CXCL12 regarding the behavior and ability of HSCs to generate B cell progenitors. HSCs migrated to CXCL12-intact CAR cells within bone marrow in Ebf3-CreERT2;CXCL12[f/-] mice, in which CXCL12 was deleted in about half of the CAR cells. In Ebf3-CreERT2;CXCL12[f/-] mice, in which CXCL12 was deleted in almost all the CAR cells, HSC numbers were reduced and the ability of HSCs to generate B cell progenitors was markedly reduced. It was reported that HSCs comprise several populations, including lymphoid-biased or balanced and myeloid-biased subsets of HSCs[19–24]. Thus, our findings suggest that CAR cell-derived CXCL12 supports the localization of HSCs and the maintenance of HSCs, especially lymphoid-biased HSCs to produce the required number of B cell progenitors.

## Results

### CXCL12 attracts HSCs to CAR cells within bone marrow

We generated CXCL12[f/-] mice (Supplementary Fig. 1) and crossed them with mice expressing the CreERT2 transgene under the control of the Ebf3 gene, in which Cre recombinase can be activated in CAR cells but not in other bone marrow cell populations when injected with tamoxifen (Ebf3-CreERT2 mice)[10]. CXCL12 was deleted in a portion of CAR cells when Ebf3-CreERT2;CXCL12[f/-] mice were injected with tamoxifen once and analyzed 10–14 weeks after tamoxifen treatment. Quantitative real-time polymerase chain reaction with reverse transcription (qRT-PCR) analysis of sorted PDGFRβ[+]Sca-1[-]CD31[-]CD45[-]Ter119[-] CAR cells[9] revealed that CXCL12 was deleted from about 70% of the CAR cells in the mutant mice (Fig. 1a). Flow cytometric analysis revealed that the bone marrow from the mutant mice contained normal total hematopoietic cell counts, normal numbers of c-kit[+]CD19[+]IgM[-] pro-B cells and c-kit[-]CD19[+]IgM[-] pre-B cells, and only modestly reduced numbers of the CD34[-]CD150[+]CD48[-] or CD150[+]CD48[-] subset of Lin[-]Sca-1[+]c-kit[+] (LSK) cells, which are highly enriched for long-term repopulating HSCs (LT-HSCs) compared with control animals (Fig. 1b–d). These results raise the possibility that CXCL12 production from a subset of CXCL12-intact CAR cells contributes to HSC maintenance in the mutants.

To address this possibility, we analyzed the localization of HSCs relative to CXCL12-intact CAR cells in these HSC-intact CXCL12 conditionally deficient mice. To visualize HSCs, we generated HSC-reporter mice, in which three EGFP genes were knocked into the HSC-specific Evi1 gene (Evi1-GFP mice) (Supplementary Fig. 2a). Evi1 is a transcription factor, which is specifically expressed in HSCs in the hematopoietic system and essential for the maintenance of HSCs[25,26]. In these mice, the majority of phenotypic and functional HSCs were Evi1-GFP[hi]c-kit[+] (Supplementary Fig. 2b, c). Conversely, the majority of Evi1-GFP[hi]c-kit[+] cells comprised CD150[+]CD48[-] LSK (LSK-SLAM) HSCs (Supplementary Fig. 2d). Limiting dilution analysis revealed that the frequency of long-term in vivo competitive repopulating units (CRU) of the Evi1-GFP[hi]c-kit[+] cells was 1 in 3.8 cells (Supplementary Fig. 2e). Flow cytometric analysis revealed that Lin[-]Sca-1[+]c-kit[+]CD150[-]CD48[-]

multipotent progenitors (MPPs) expressed lower levels of Evi1-GFP compared to HSCs and other primitive hematopoietic progenitors, including Lin[-]Sca-1[-]c-kit[+]CD34[+]FcγRII/III[hi] granulocyte/macrophage progenitors (GMPs) and Lin[-]Sca-1[-]c-kit[+]CD34[-]FcγRII/III[lo] mega-karyocyte/erythrocyte progenitors (MEPs), did not express Evi1-GFP (Supplementary Fig. 2b). We could detect the fluorescence signals in fixed HSCs but not in fixed MPPs isolated from Evi1-GFP mice by histology (Supplementary Fig. 3). Flow cytometric analysis revealed HSC numbers were unaltered in Evi1-GFP mice (Supplementary Fig. 4). In addition, the mRNA levels of Evi1 in HSCs were unaltered in the absence of CXCL12 in CAR cells as described later. Together, Evi1-GFP mice allowed visualization of HSCs in the bone marrow sections of control and CXCL12 conditionally deficient mice. To visualize CXCL12-intact and CXCL12-deficient CAR cells, we generated another conditional CXCL12-targeted mouse line by inserting loxP sites that flank exons 2 and 3 of the Cxcl12 gene and a linked tandem dimer Tomato (tdTomato) gene to act as a CXCL12-specific reporter (CXCL12-tdTomato[f/f] mice) (Fig. 1e), and we crossed them with Ebf3-CreERT2 mice. Control CXCL12-tdTomato[f/f] and Ebf3-CreERT2;CXCL12-tdTomato[f/f] mice were transplanted with bone marrow cells from Evi1-GFP HSC-reporter mice, injected with tamoxifen three times after 16 weeks following transplantation, and analyzed 17 to 23 weeks after tamoxifen treatment. In these chimeric mice, the red fluorescent protein tdTomato was expressed in CXCL12-intact CAR cells but not in CXCL12-deficient CAR cells. Immunohistochemical analysis with antibodies against S100, which is specifically expressed in CAR cells in bone marrow cavities[9], showed that about 42% of the S100[+] CAR cells lacked CXCL12-tdTomato expression in mutant chimeric mice, indicating that CXCL12-intact CAR cells were reduced by about 2-fold (Fig. 1f). Consistent with this, frequencies of random spots located within 5 μm of CXCL12-intact CAR cells were reduced compared with control chimeras (Fig. 1g). However, frequencies and numbers of HSCs were unaltered (Fig. 1h and Supplementary Fig. 5) and frequencies of Evi1-GFP[hi]c-kit[+] HSCs located within 5 μm distance from CXCL12-intact CAR cells were unaltered in the mutant chimeras (Fig. 1i, j and Supplementary Fig. 6a). We next compared the distribution of HSCs with the distribution of CXCL12-intact and CXCL12-deficient CAR cells using antibodies against PDGFRβ, which is preferentially expressed in CAR cells in the bone marrow. Although the numbers of CXCL12-intact and CXCL12-deficient CAR cells were similar (Fig. 1f), frequencies of HSCs located within 5 μm distance from CXCL12-tdTomato[-]PDGFRβ[+] CXCL12-deficient CAR cells were markedly reduced compared to those from CXCL12-tdTomato[+]PDGFRβ[+] CXCL12-intact CAR cells in the mutant chimeras (Fig. 1k, l and Supplementary Fig. 6b). These results reveal that HSCs detached from CXCL12-deficient CAR cells and attached to CXCL12-intact CAR cells in the mutants, indicating that CXCL12 attracts HSCs to CAR cells within bone marrow. On the other hand, few HSCs are located within 5 μm distance from the bone surface, suggesting that few HSCs were maintained in the endosteal niches in the mutants.

### CAR/LepR[+] cell-derived CXCL12 is essential for the maintenance of HSCs in aged mice

Because CXCL12 production from a small subset of CAR cells might contribute to HSC maintenance in 16-week-old LepR-Cre;CXCL12[f/-] mice, we examined HSCs and their progeny in 90-week-old LepR-Cre;CXCL12[f/-] mice, in which LepR was expressed in almost all the CAR cells[4]. qRT-PCR analysis of sorted PDGFRβ[+]Sca-1[-]CD31[-]CD45[-]Ter119[-] CAR cells revealed that CXCL12 was deleted in about 99.5% of the CAR cells (Fig. 2a). Flow cytometric analysis revealed that the total hematopoietic cell counts and numbers of LT-HSCs, Lin[-]IL-7Rα[+]Flt3[+] common lymphoid progenitors (CLPs), pro-B cells, pre-B cells, mature B cells, pDCs, NK cells, GMPs, Gr-1[hi]CD11b[+] granulocytes, MEPs, and c-kit[+]CD71[+]Ter119[lo] proerythroblasts were severely reduced in the bone marrow of the mutant mice compared with control animals (Fig. 2b–d).

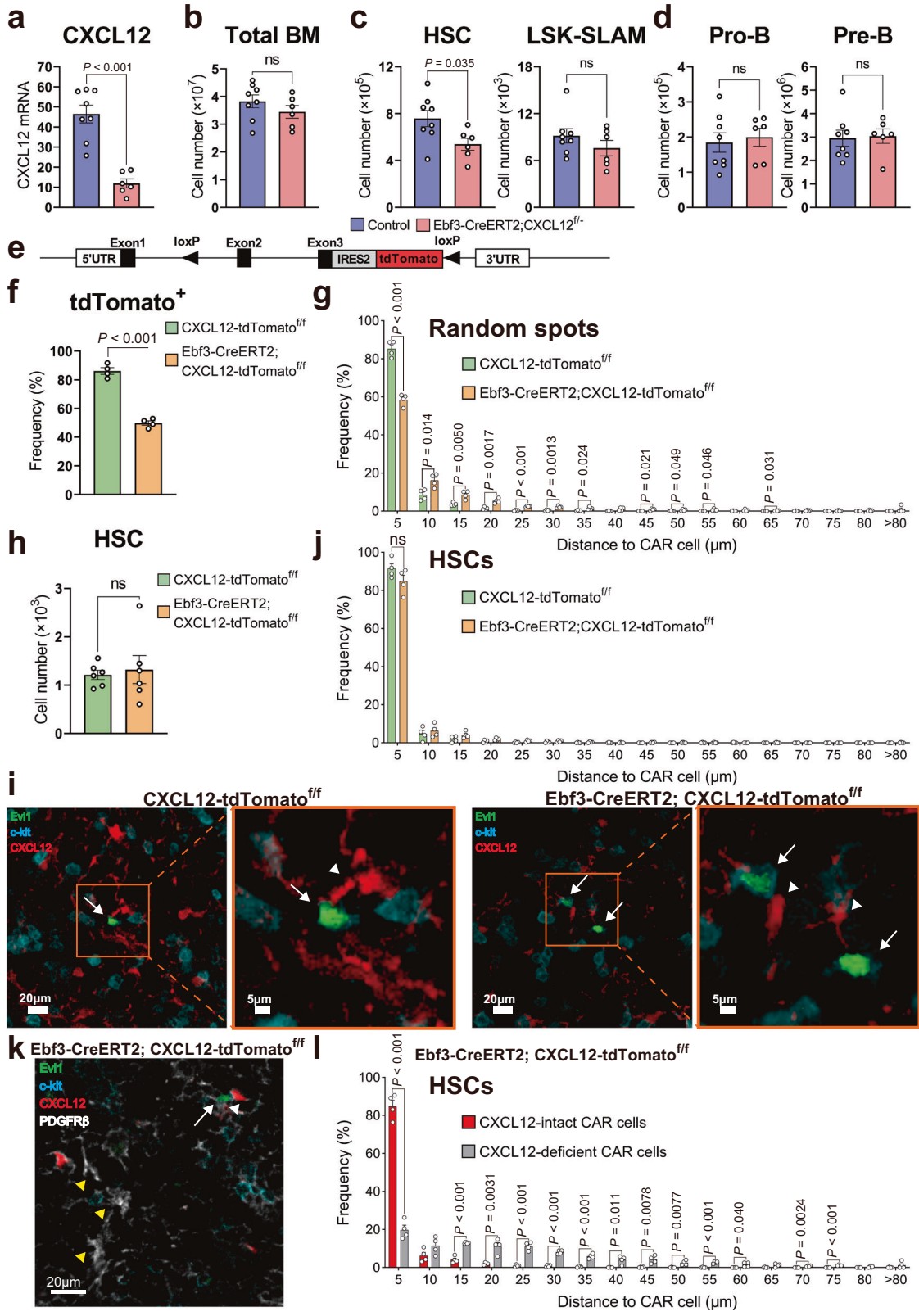

The magnitude of the reduction was increased in B cell progenitors compared with LT-HSCs, GMPs, and MEPs in aged LepR-Cre;CXCL12$^{f/-}$ mice. Consistent with this, frequencies of B cell progenitors were severely reduced in the bone marrow of the mutant mice compared with control animals (Supplementary Fig. 7b). In addition, we estimated the repopulating potential in HSCs using repopulating units (RUs),

based on a competitive repopulation assay, where short-lived peripheral blood myeloid cells were analyzed, and found that RUs were markedly reduced in bone marrow of the mutants (Fig. 2e). These results indicate that CAR/LepR$^+$ cell-derived CXCL12 is essential for the maintenance of HSCs and hematopoietic progenitors in the bone marrow of aged mice.

**Fig. 1 | CXCL12 attracts HSCs to CAR cells. a–d** Bone marrow from 21- to 25-week-old Ebf3-CreERT2;CXCL12[+/+] control (n = 8) and Ebf3-CreERT2;CXCL12[f/-] conditional knockout mice (n = 6) injected once with tamoxifen was analyzed. CXCL12 mRNA levels in CAR cells (**a**), the total hematopoietic cell counts (**b**), and the numbers of CD34[-]CD150[+]CD48[-]LSK (HSCs), CD150[+]CD48[-]LSK (LSK-SLAM) HSCs (**c**), pro-B cells and pre-B cells (**d**) in the bone marrow. **e** Schematic illustrating generation of the CXCL12-tdTomato[f/f] mouse line. **f–l** CXCL12-tdTomato[f/f] control and Ebf3-CreERT2;CXCL12-tdTomato[f/f] conditional knockout mice transplanted with bone marrow cells from Evi1-GFP mice were injected with tamoxifen three times and analyzed 17 to 23 weeks after treatment. Frequencies of CXCL12-tdTomato[+] cells in S100[+] CAR cells (**f**) and frequencies of random spots within each distance from CXCL12-intact CAR cells (**g**) in the bone marrow sections (n = 4). The numbers of phenotypic LT-HSCs in the bone marrow (**h**) (n = 6). Immunohistochemical analysis of Evi1-GFP[hi]c-kit[+] HSCs (arrow) in contact with CXCL12-intact CAR cells (arrowhead) (**i**). The frequencies of Evi1-GFP[hi]c-kit[+] HSCs within each distance from CXCL12-intact CAR cells in the bone marrow sections (n = 4) (**j**). Immunohistochemical analysis of Evi1-GFP[hi]c-kit[+] HSCs (white arrows) in contact with the CXCL12-tdTomato[+]PDGFRβ[+] CXCL12-intact CAR cells (white arrowheads) and distant from the CXCL12-tdTomato[-]PDGFRβ[+] CXCL12-deficient CAR cells (yellow arrowheads) in the mutants (**k**). The frequencies of Evi1-GFP[hi]c-kit[+] HSCs within each distance from CXCL12-tdTomato[+]PDGFRβ[+] CXCL12-intact CAR cells or CXCL12-tdTomato[-]PDGFRβ[+] CXCL12-deficient CAR cells in the bone marrow sections of the mutants (n = 4) (**l**). In (**j**), total of 960 and 1030 Evi1-GFP[hi]c-kit[+] HSCs in 50 and 52 bone marrow sections were analyzed in CXCL12-tdTomato[f/f] control and Ebf3-CreERT2;CXCL12[f/-] conditional knockout mice, respectively, in 4 independent experiments. In (**l**), total 789 Evi1-GFP[hi]c-kit[+] HSCs in 36 bone marrow sections were analyzed in 4 independent experiments. All error bars represent SE of the mean. Statistical significances were calculated using the two-tailed unpaired Student's t test. Source data are provided as a Source data file.

## CAR cell-derived CXCL12 is essential for the maintenance of HSCs and hematopoietic progenitors

To examine the role of CXCL12 produced by CAR cells in younger adults, Ebf3-CreERT2;CXCL12[f/-] mice were injected with tamoxifen eight times every other day beginning at 10 weeks of age and analyzed at 10 to 14 weeks after tamoxifen treatment. qRT-PCR analysis of sorted PDGFRβ[+]Sca-1[-]CD31[-]CD45[-]Ter119[-] CAR cells revealed that CXCL12 was deleted from more than 99.5% of the CAR cells in the majority of tamoxifen-treated Ebf3-CreERT2;CXCL12[f/-] mice (CXCL12[ΔCAR] mice) (Fig. 3a). Flow cytometric analysis of CXCL12[ΔCAR] mice revealed that the total hematopoietic cell counts and frequencies and numbers of LT-HSCs, EPCR[+]CD150[+]CD48[-]LSK (LSK-ESLAM) HSCs, LSK-SLAM HSCs, MPPs, CLPs, pro-B cells, pre-B cells, mature B cells, pDCs, NK cells, GMPs, granulocytes, MEPs, and proerythroblasts were reduced in the bone marrow of the mutants compared with control animals (Fig. 3b–e and Supplementary Fig. 8a–c). Of note, the magnitude of the reduction was increased in B cell progenitors compared with LT-HSCs, GMPs, and MEPs in the bone marrow of CXCL12[ΔCAR] mice. In addition, the numbers of Lin[-]Sca-1[+]c-kit[+]CD150[-]CD48[-]flt3[+] MPP4s with high lymphoid and low megakaryocyte/erythroid potential were reduced; however, the numbers of Lin[-]Sca-1[+]c-kit[+]CD150[-]CD48[+]flt3[-] MPP3s with high myeloid potential were unaltered and Lin[-]Sca-1[+]c-kit[+]CD150[+]CD48[+]flt3[-] MPP2s with low lymphoid and high megakaryocyte/erythroid potential[27] were modestly increased (Fig. 3d and Supplementary Fig. 8b). We next analyzed RUs and the frequency and number of functional HSCs, which were determined using CRUs based on limiting-dilution of competitive repopulation assays, where short-lived peripheral blood myeloid cells were analyzed. The numbers of CRUs and RUs were markedly reduced in the bone marrow of CXCL12[ΔCAR] mice (Fig. 3f). To confirm this, we performed secondary transplantations. Mice competitively reconstituted with 1500 donor-derived Lin[-]Sca-1[+]c-kit[+] primitive hematopoietic stem and progenitor cells from primary recipients transplanted with 200 cells in the HSC population from CXCL12[ΔCAR] mice exhibited markedly reduced long-term multilineage reconstitution by donor-derived HSCs in secondary recipients (Supplementary Fig. 9). Cyclin-dependent kinase inhibitor p57 is essential for quiescence and maintenance of HSCs[28]. The mRNA levels of p57 were reduced but those of the transcription factor Evi1 essential for HSC maintenance were unaltered in LT-HSCs in the marrow of CXCL12[ΔCAR] mice (Fig. 3g). These results indicate that CAR cell-derived CXCL12 is essential for the maintenance of HSCs and hematopoietic progenitors within bone marrow.

## The ability of HSCs to generate B cell progenitors was markedly reduced in mice lacking CXCL12 in CAR cells

Since it was previously demonstrated that distinct HSCs exist that are stably biased towards the generation of lymphoid or myeloid cells[19–24], we examined the ability of HSCs from CXCL12[ΔCAR] and control mice to generate B cell or myeloid progenitors within wild-type bone marrow.

Sorted 200 cells in the HSC population from CXCL12[ΔCAR] or control mice were transplanted with 5×10[5] competitor bone marrow cells into wild-type mice. The percentages of donor-derived LT-HSCs, pro-B cells, pre-B cells, immature B cells, mature B cells, GMPs, and MEPs in the bone marrow were analyzed at 16 weeks after transplantation (Fig. 4a, c–e and Supplementary Fig. 10a–c) and the percentages of donor-derived B220[+] B cells and Gr-1[hi] granulocytes in the peripheral blood were analyzed at 4, 8, 12, 16 weeks after transplantation (Fig. 4b, f and Supplementary Fig. 9 and 10b, c). In the bone marrow, donor B and myeloid cell chimerism was reduced in recipients of HSCs from CXCL12[ΔCAR] mice (Fig. 4a and Supplementary Fig. 10a), but donor contribution into B cell progenitors, i.e. % donor B cell progenitors divided by % donor myeloid progenitors (donor pro-B/GMP and pre-B/GMP reconstitution ratios), in recipients of HSCs from CXCL12[ΔCAR] mice was markedly lower than that in recipients of HSCs from control animals (Fig. 4c), indicating a stable myeloid bias of HSCs from CXCL12[ΔCAR] mice. In addition, the donor B cell progenitor/erythro-megakaryocytic progenitor (pro-B/MEP and pre-B/MEP) reconstitution ratios (Fig. 4d) and % donor B cell progenitors divided by % donor LSK-SLAM HSCs (pro-B/LSK-SLAM and pre-B/LSK-SLAM) in recipients of HSCs from CXCL12[ΔCAR] mice (Fig. 4e) were markedly lower than those in recipients of HSCs from control animals. Consistent with this, the donor B cell/granulocyte reconstitution ratios in the peripheral blood were lower in recipients of HSCs from CXCL12[ΔCAR] mice than those in recipients of HSCs from control animals (Fig. 4f). These results confirmed the attenuated ability of HSCs from CXCL12[ΔCAR] mice to generate B lineage progeny. Consistent with this, frequencies of the CD150[lo] or CD229[hi] subset, which gave a higher level of lymphoid reconstitution[23,29], in the LT-HSC population were reduced in CXCL12[ΔCAR] mice (Fig. 4g and Supplementary Fig. 11). Interestingly, the magnitudes of the decreases in the donor B cell progenitor/myeloid progenitor and B cell progenitor/erythro-megakaryocytic progenitor reconstitution ratios in mutant HSC chimeric mice were not smaller when compared to ratios of B cell progenitor numbers to myeloid or erythro-megakaryocytic progenitor numbers in CXCL12[ΔCAR] mice (Compare Fig. 4c, d with 4h). Similar results were obtained when we performed secondary transplantations. We transplanted 1500 donor-derived Lin[-]Sca-1[+]c-kit[+] primitive hematopoietic stem and progenitor cells from primary recipients transplanted with 200 cells in the HSC population. The ability of HSCs from CXCL12[ΔCAR] mice to generate B cell progenitors was markedly reduced after transplantation in secondary recipients (Supplementary Fig. 12).

## CXCL12 enabled the maintenance of the B lineage repopulating ability of HSCs in vitro

The preferential reduction in lymphoid-biased HSCs in the absence of CAR cell-derived CXCL12 prompted us to examine direct actions of CXCL12 on maintenance of the lymphoid repopulating ability of HSCs. A serum-free culture system that supports the functional mouse HSCs

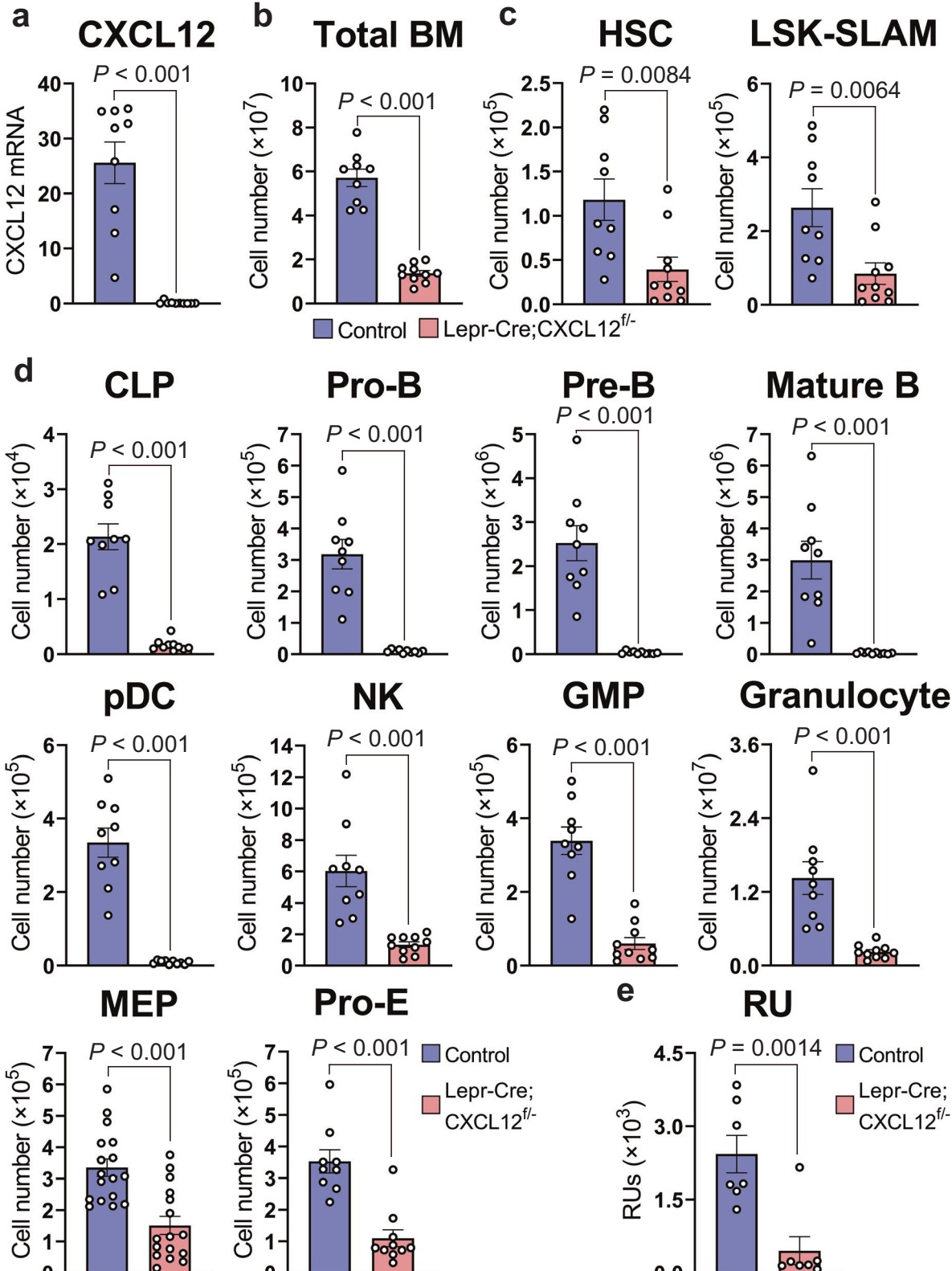

**Fig. 2 | LepR⁺ cells-derived CXCL12 is essential for HSC maintenance and B cell progenitor production in aged mice. a** CXCL12 mRNA levels in CAR cells from 90-week-old LepR-Cre;CXCL12$^{+/+}$ control ($n = 9$) and LepR-Cre;CXCL12$^{f/-}$ conditional knockout ($n = 10$) mice. **b–e** The total hematopoietic cell counts (**b**) and the numbers of CD34⁻CD150⁺CD48⁻LSK HSCs, LSK-SLAM HSCs (**c**), CLPs, pro-B cells, pre-B cells, mature B cells, pDCs, NK cells, GMPs, granulocytes, MEPs, proerythroblasts (pro-E) (**d**), and repopulating units (RUs) (**e**) in the bone marrow from 90-week-old LepR-Cre;CXCL12$^{+/+}$ control ($n = 9$) and LepR-Cre;CXCL12$^{f/-}$ conditional knockout ($n = 10$) mice. All error bars represent SE of the mean. Statistical significances were calculated using the two-tailed unpaired Student's *t* test. Source data are provided as a Source data file.

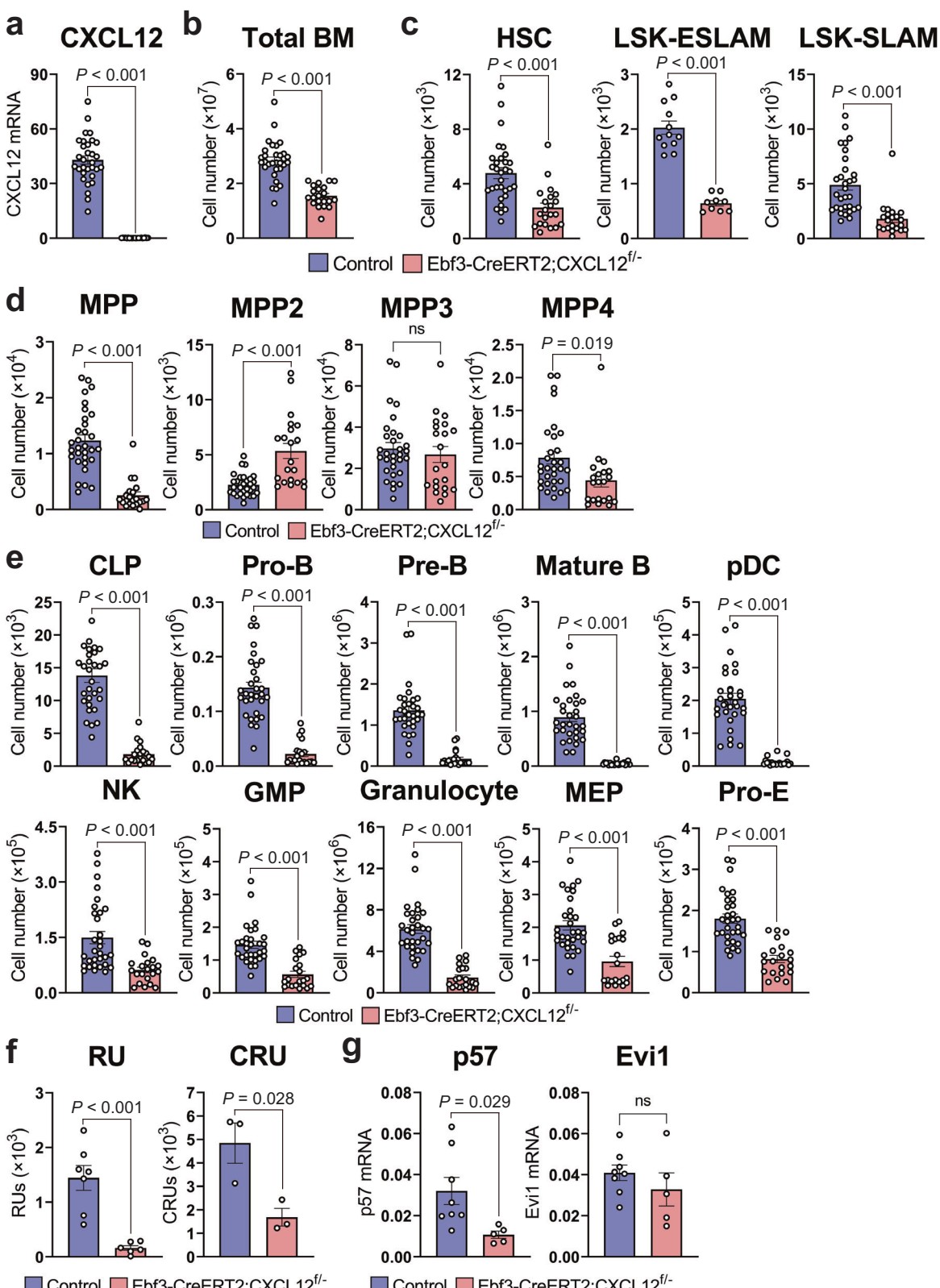

ex vivo for over one month has been developed[30]. Using this culture system, we sorted 50 cells in the LT-HSC population into fibronectin-coated plate wells containing serum-free medium and polyvinyl alcohol (PVA), cultured them for 28 days in the presence of stem cell factor (SCF) and thrombopoietin (TPO) with or without CXCL12, and transplanted cultured cells with $1 \times 10^6$ CD45.1$^+$ competitor bone marrow cells into wild-type mice (Fig. 5a). At 16 weeks after transplantation,

although percentages of donor-derived hematopoietic cells were variable between recipients (Fig. 5b and Supplementary Fig. 13), the donor B cell progenitor/myeloid progenitor (pro-B/GMP and pre-B/GMP) reconstitution ratios in recipients of cultured cells without CXCL12 was lower than the donor B cell progenitor/myeloid progenitor reconstitution ratios in recipients of HSCs from wild-type mice (compare Fig. 4c with Fig. 4d). However, interestingly, the donor B cell

**Fig. 3 | CAR cell-derived CXCL12 is essential for the HSC maintenance and production of B cell progenitors.** Bone marrow from 21- to 25-week-old Ebf3-CreERT2;CXCL12+/+ control and Ebf3-CreERT2;CXCL12f/- CXCL12ΔCAR mice injected with tamoxifen eight times was analyzed. **a**−**e** CXCL12 mRNA levels in CAR cells (**a**), the total hematopoietic cell counts (**b**), numbers of CD34-CD150+CD48-LSK HSCs, LSK-ESLAM HSCs, LSK-SLAM HSCs (**c**), MPPs, MPP2s, MPP3s, MPP4s (**d**), CLPs, pro-B cells, pre-B cells, mature B cells, pDCs, NK cells, GMPs, granulocytes, MEPs and proerythroblasts (pro-E) (**e**) in the bone marrow of Ebf3-CreERT2;CXCL12+/+ control ($n = 31$) and CXCL12ΔCAR ($n = 21$) mice. **f** RUs of Ebf3-CreERT2;CXCL12+/+ control ($n = 7$) and CXCL12ΔCAR ($n = 6$) mice and the numbers of functional LT-HSCs in the bone marrow, which were determined by measuring the competitive repopulating units (CRUs) ($n = 3$). **g** The mRNA levels of p57 and Evi1 in LT-HSCs in the bone marrow of Ebf3-CreERT2;CXCL12+/+ control ($n = 8$) and CXCL12ΔCAR ($n = 5$) mice. All error bars represent SE of the mean. Statistical significances were calculated using the two-tailed unpaired Student's $t$ test. Source data are provided as a Source data file.

progenitor/myeloid progenitor (pro-B/GMP and pre-B/GMP) reconstitution ratios in the bone marrow and B cell/myeloid cell reconstitution ratios in the peripheral blood in recipients of cultured cells with CXCL12 was higher than those in recipients of cultured cells without CXCL12 (Fig. 5c, d) and was comparable to those in recipients of HSCs from wild-type mice (compare Fig. 4c with Fig. 4d).

## Discussion

It was reported that the number of LT-HSCs was unaltered in bone marrow from young LepR-Cre;CXCL12f/- mice, in which CXCL12 was deleted in about 70% of the CAR cells[16,17]. In this study, we have shown that most HSCs were in contact with intact CAR cells in the mutants, in which CXCL12 was deleted in about half of the CAR cells. This result suggests that when CXCL12 was deleted from some CAR cells, HSCs became detached from these CXCL12-deficient CAR cells and migrated to CXCL12-intact CAR cells and were maintained. Our results are compelling evidence that CXCL12 attracts HSCs to CAR cells within bone marrow.

In contrast to young LepR-Cre;CXCL12f/- mice, aged LepR-Cre;CXCL12f/- mice and CXCL12ΔCAR mice, in which CXCL12 was deleted from more than 99.5% of the CAR cells, had markedly reduced numbers of functional HSCs compared with control animals. Although it was reported that LepR+ cell-derived CXCL12 is essential to inhibit egress of HSCs from bone marrow but not to maintain HSCs[16], our results strongly suggest that CAR cell-derived CXCL12 is essential for the maintenance of HSCs. Our results cannot exclude the possibility that CXCL12 produced outside the bone marrow and/or by progeny of CAR cells, including osteoblasts contributes to HSC maintenance. However, the facts that bone marrow is one of the largest organs and the expression levels of CXCL12 in CAR cells abundant in the marrow are much higher than any other type of cells in the body suggest that CAR cells are a major source of systemic CXCL12 levels and support our conclusion.

We considered how CXCL12 maintains HSCs and provides the proper number of B cell progenitors. If CXCL12 acts on HSCs to increase their ability to generate B cell progenitors or induce differentiation of HSCs into B cell progenitors and/or acts on lineage-committed progenitors, then the donor B lymphoid/myeloid reconstitution ratios in wild-type recipients transplanted with HSCs from CXCL12ΔCAR mice become comparable to those in wild-type recipients transplanted with HSCs from wild-type mice. However, even when HSCs from CXCL12ΔCAR mice were transplanted into wild-type recipient mice, the donor B lymphoid/myeloid reconstitution ratios in the recipients were markedly lower than those in recipients of HSCs from control animals for 16 weeks, indicating that the ability of HSCs to generate B cell progenitors was reduced, and the reduction was irreversible in CXCL12ΔCAR mice. It has been shown previously that HSCs display heterogeneity in their differentiation potential containing distinct HSC subsets that are stably biased towards the generation of lymphoid or myeloid blood cells[19–23]. Thus, our results suggest that CAR cell-derived CXCL12 preferentially acts on lymphoid-biased HSCs with a stable B cell repopulation ability to self-renew. We cannot, however, exclude the possibility that continued reduction of HSCs increased cell division of lymphoid-biased HSCs in an effort to repair the hematopoietic defects, resulting in reduced generation of B cell

progenitors from remaining lymphoid-biased HSCs. On the other hand, previous studies have shown the existence of balanced/multipotent HSCs and platelet-biased HSCs but not lymphoid-biased HSCs[24]. In this case, balanced/multipotent HSCs were markedly reduced and remaining HSCs would be biased toward the generation of myeloid progenitors in CXCL12ΔCAR mice. Thus, it is not clear whether the stronger phenotype in the lymphoid lineage reflects a direct effect of CXCL12 on lymphoid-biased or balanced HSCs versus an indirect effect of depleted HSC self-renewal potential after increased cell division.

In addition to the donor B lymphoid/myeloid reconstitution ratios, the donor B lymphoid/HSC reconstitution ratios were markedly reduced in wild-type recipients transplanted with HSCs from CXCL12ΔCAR mice. These results suggest that frequencies of HSCs, which generated B cell progenitors were reduced and/or the ability of HSCs to generate B cell progenitors was reduced to a larger extent than the ability of HSCs to self-renew in CXCL12ΔCAR mice compared with wild-type mice.

The magnitude of the decrease in the donor pre-B/GMP reconstitution ratios in mutant chimeric mice transplanted with HSCs from CXCL12ΔCAR mice was similar compared with pre-B/GMP ratios in CXCL12ΔCAR mice (Fig. 4). Moreover, in mice whose multipotent hematopoietic progenitors and their progeny lack the CXCL12 receptor, CXCR4, the numbers of pre-B cells were only modestly reduced[11]. These results suggest that a markedly smaller number of B cell progenitors was mainly due to the preferential reduction in lymphoid-biased or balanced HSCs in CXCL12ΔCAR mice. Niche-derived IL-7 is known to act on CLPs and B cell progenitors to enhance their differentiation and proliferation[11]. Thus, this study establishes an additional mechanism by which niche-derived cytokines regulate B cell development.

LepR-Cre;SCFf/- mice lacking SCF in the majority of CAR cells have markedly reduced numbers of HSCs[8] but have normal numbers of pro-B cells, pre-B cell, and mature B cells[31] in contrast to CXCL12ΔCAR mice, suggesting that the specific function of maintaining lymphoid-biased HSCs is possessed by CXCL12 but not SCF. Therefore, it is unlikely that CXCL12 merely attracts HSCs to CAR cells to receive the CAR cell-surface molecules, including membrane-bound SCF.

Recently, a serum-free culture system that supports the functional mouse HSCs ex vivo for over one month has been developed[30]. However, we have revealed that cultured HSCs showed a myeloid bias compared with HSCs in the bone marrow. Since lineage dominance is a stable property of HSCs, it is important to provide culture conditions that substitute for the complex microenvironment of intact bone marrow to maintain lymphoid-biased HSCs for long-term ex vivo expansion of HSCs. Thus, our findings that CXCL12 enabled the maintenance of the B lineage repopulating ability of HSCs in vitro confirmed direct actions of CXCL12 on the self-renewal and/or survival of lymphoid-biased HSCs in addition to their attachment to CAR cells. These results raise the possibility that CXCL12 or its activators may be clinically useful for the long-term ex vivo expansion of HSCs with normal lymphoid potential. Altogether, this study provides insight of cellular and molecular mechanisms by which HSC niches act on HSCs to regulate B cell development.

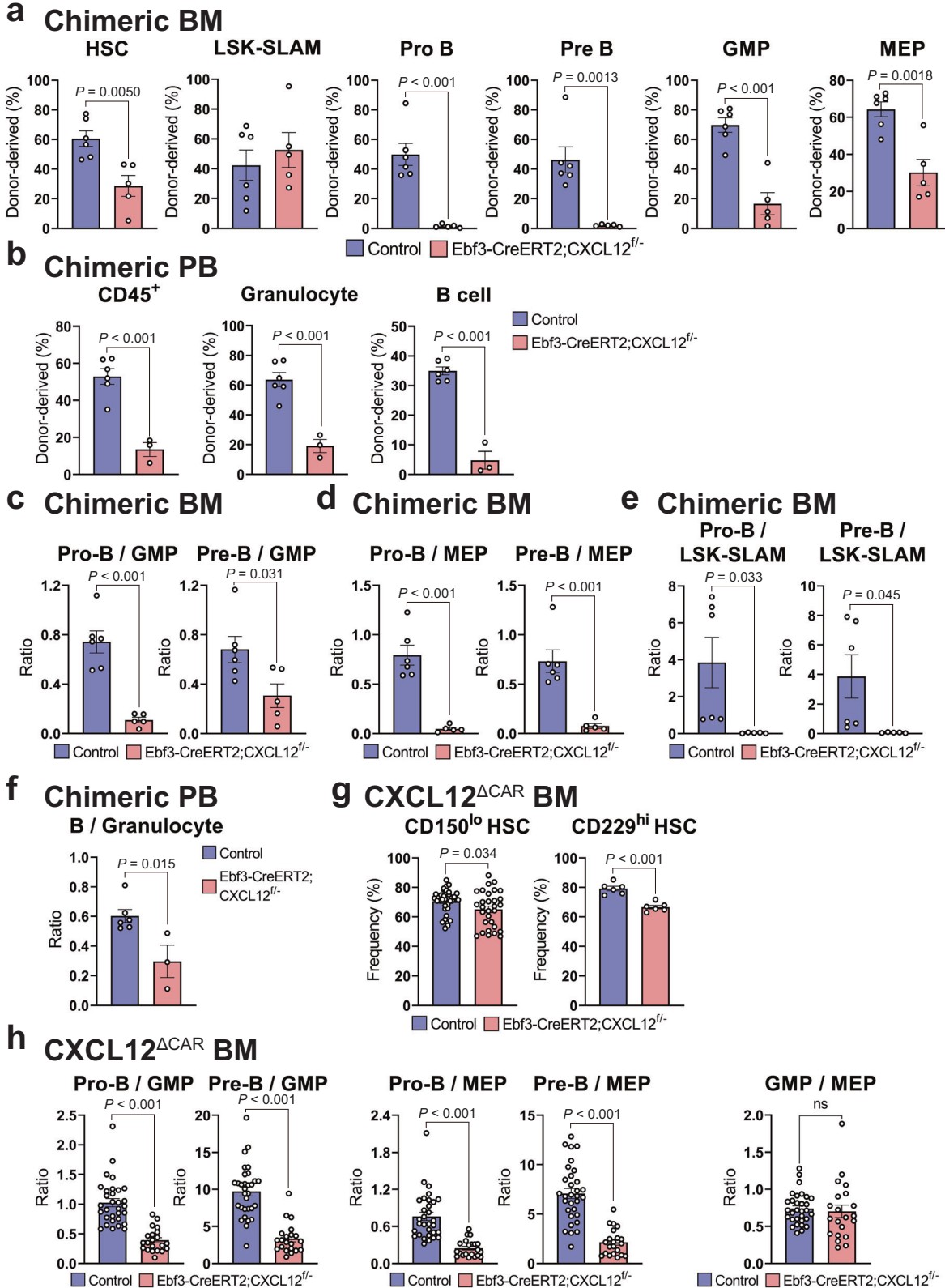

## Methods

### Mice

Targeting vectors for making CXCL12$^{f/f}$ and CXCL12-tdTomato$^{f/f}$ mice were constructed as shown (Fig. 1c and Supplementary Fig. 1). The FRT-flanked Neo cassette was removed by mating the Flpe mice[32]. These mice were backcrossed at least seven times onto a C57BL/6 background before analysis. Targeting vector and gRNA for making Evi1-GFP mice were constructed as shown in Supplementary Fig. 2a. CXCL12$^{+/-}$ and Ebf3-CreERT2 mice have been described previously[10,32,33]. Lepr-Cre mice[34] were obtained from the Jackson Laboratory. All mice were maintained on a C57BL/6 background. To induce CreERT2-mediated recombination, Ebf3-CreERT2;CXCL12$^{f/-}$ mice were injected intraperitoneally with 2 mg of tamoxifen (Cayman Medical) 1 or 8 times every other day, and Ebf3-CreERT2;CXCL12-tdTomato$^{f/f}$ mice were injected 3

**Fig. 4 | The ability of HSCs to generate B cell progenitors was markedly reduced in mice lacking CXCL12 in CAR cells. a–f** Sorted 200 cells in the HSC population from 21- to 25-week-old tamoxifen-treated CD45.2⁺ Ebf3-CreERT2;CXCL12⁺/⁺ control or Ebf3-CreERT2;CXCL12ᶠ/⁻ CXCL12ᐞCAR mice, in which CXCL12 was deleted from more than 99.5% of the CAR cells (CXCL12ᐞCAR mice) were transplanted with $5 \times 10^5$ CD45.1⁺ competitor bone marrow cells into CD45.1⁺CD45.2⁺ wild-type mice. The percentages of donor-derived cells were analyzed at 16 weeks after transplantation. **a, b** Donor chimerism of hematopoietic stem and progenitor cells in the bone marrow of recipients transplanted with HSCs from Ebf3-CreERT2;CXCL12⁺/⁺ control (n = 6) or CXCL12ᐞCAR (n = 5) mice (**a**) and CD45⁺ cells, granulocytes, and B cells in the peripheral blood of recipients transplanted with HSCs from Ebf3-CreERT2;CXCL12⁺/⁺ control (n = 6) or CXCL12ᐞCAR (n = 3) mice (**b**). **c–e** % donor B cell progenitors divided by % donor myeloid progenitor (donor pro-B/GMP and pre-B/GMP reconstitution ratios) (**c**), % donor B cell progenitors divided by % donor erythro-megakaryocytic progenitors (donor pro-B/MEP and pre-B/MEP reconstitution ratios) (**d**), and % donor B cell progenitors divided by % donor LSK-SLAM HSCs (donor pro-B/LSK-SLAM and pre-B/LSK-SLAM reconstitution ratios) (**e**) in the bone marrow of recipients transplanted with HSCs from Ebf3-CreERT2;CXCL12⁺/⁺ control (n = 6) or CXCL12ᐞCAR (n = 5) mice are shown. **f** % donor B cells divided by % donor granulocytes (donor B/granulocyte reconstitution ratios) in the peripheral blood of recipients transplanted with HSCs from Ebf3-CreERT2;CXCL12⁺/⁺ control (n = 6) or CXCL12ᐞCAR (n = 3) mice are shown. **g** Frequencies of the CD150ˡᵒ subset in the LT-HSC population in 21- to 25-week-old Ebf3-CreERT2;CXCL12⁺/⁺ control (n = 37) and CXCL12ᐞCAR (n = 30) mice or CD229ʰⁱ subset in the LT-HSC population in 21- to 25-week-old Ebf3-CreERT2;CXCL12⁺/⁺ control and CXCL12ᐞCAR (n = 6) mice. **h** Ratios of B cell progenitor numbers to myeloid progenitor numbers (pro-B/GMP and pre-B/GMP) or erythro-megakaryocytic progenitor numbers (pro-B/MEP and pre-B/MEP) and ratio of myeloid progenitor numbers to erythro-megakaryocytic progenitor numbers (GMP/MEP) in 21- to 25-week-old Ebf3-CreERT2;CXCL12⁺/⁺ control (n = 31) and CXCL12ᐞCAR (n = 21) mice. All error bars represent SE of the mean. Statistical significances were calculated using the two-tailed unpaired Student's t test. Source data are provided as a Source data file.

times every other day and analyzed 10–23 weeks after the final tamoxifen injection. All animal experiments were performed in accordance with approved protocols of the Institutional Animal Care and Use Committees at Osaka University.

## Flow cytometric analysis and cell sorting

All flow cytometric experiments and cell sorting were performed using a BD FACSAria (BD Biosystems). For sorting the nonhematopoietic cell populations, cells in bone marrow fraction were obtained from femurs and tibiae by flushing and collagenase (Sigma) digestion. Lineage markers used for hematopoietic stem and progenitor analysis were Ter119, CD3, B220, Gr-1, and CD11b. Gating strategies are included in Supplementary Fig. 14. The antibodies used for flow cytometric analysis are listed in Supplementary Table 1 in Supplementary Information file.

## Competitive repopulation assay

200 CD150⁺CD48⁻LSK cells were sorted from femurs and tibiae from Ebf3-CreERT2;CXCL12ᶠ/⁻ mice (C57BL/6-CD45.2⁺ background), and 200 Evi1-GFP⁺CD150⁺CD48⁻LSK or Evi1-GFP⁻CD150⁺CD48⁻LSK cells were sorted from Evi1-GFP mice (C57BL/6-CD45.2⁺ background). Cells to be tested were mixed with $5 \times 10^5$ bone marrow cells of 8–12-week-old C57BL/6-CD45.1⁺ mice as competitor cells and were transplanted into lethally irradiated (9 Gy) C57BL/6-CD45.1⁺CD45.2⁺ mice. For secondary transplantations, 1500 donor-derived CD45.2⁺Lin⁻Sca-1⁺c-kit⁺ primitive hematopoietic stem and progenitor cells from primary recipients were sorted. Cells to be tested were mixed with $1 \times 10^6$ bone marrow cells of 8–12-week-old C57BL/6-CD45.1⁺ mice as competitor cells and were transplanted into lethally irradiated (9 Gy) C57BL/6-CD45.1⁺CD45.2⁺ mice. Cultured CD34⁻CD150⁺CD48⁻LSK cells from C57BL/6-CD45.2⁺ mice to be tested were mixed with $5 \times 10^5$ bone marrow cells of 8 to 12-week-old C57BL/6-CD45.1⁺ mice as competitor cells and were transplanted into lethally irradiated (9 Gy) C57BL/6-CD45.1⁺CD45.2⁺ mice. Peripheral blood cells of the recipient mice were analyzed at 4, 8, 12, 16 weeks after transplantation and bone marrow cells of the recipient mice were analyzed 16 weeks after transplantation by flow cytometry. Repopulation Units (RU) were calculated using Harrison's formula as described previously[35].

## Limiting dilution analysis

For limiting dilutions of Ebf3-CreERT2;CXCL12ᶠ/⁻ mice (C57BL/6-CD45.2⁺ background), bone marrow cells to be tested were mixed with $2 \times 10^5$ bone marrow cells from 8 to 12-week-old C57BL/6-CD45.1⁺ mice and transplanted into lethally irradiated (9 Gy) C57BL/6-CD45.1⁺CD45.2⁺ mice. CRU amounts were determined in three separate experiments, by injecting 4 recipients at each cell dose in a dilution series. Cell doses

ranged from $2.5 \times 10^3$ to $2 \times 10^4$ cells. Peripheral blood cells of the recipient mice were harvested 16 weeks after transplantation, and myeloid cells were analyzed by flow cytometry. For limiting dilutions of Evi1-GFP mice (C57BL/6-CD45.2⁺ background), Evi1-GFPʰⁱc-kit⁺ cells were isolated and mixed with $5 \times 10^5$ bone marrow cells from 8 to 12-week-old C57BL/6-CD45.1⁺ mice and transplanted into lethally irradiated (8 Gy) C57BL/6-CD45.1⁺CD45.2⁺ mice. CRU amounts were determined in two separate experiments. Peripheral blood cells of the recipient mice were harvested 12 weeks after transplantation, and myeloid cells were analyzed by flow cytometry. CRU frequencies were calculated with ELDA software[36].

## Cell culture

For ex vivo culture, CD34⁻CD150⁺CD48⁻LSK cells were isolated from femurs and tibiae of C57BL/6-CD45.2⁺ mice by flow cytometry, plated into flat-bottomed 96-well fibronectin-coated plates (Corning; 354409) at 50 cells per well in medium composed of F12 medium (Wako), 1% ITSX (Gibco), 10 mM HEPES (Gibco), 1% P/S/G (Gibco), 100 ng/ml mouse TPO (Wako), 10 ng/ml mouse SCF, 0.1% PVA (Peprotech) with or without 100 ng/ml mouse CXCL12 (Kola-Gen Pharma), and cultured at 37 °C with 5% $CO_2$ for 28 days. Medium changes were performed as described previously[30]. Following ex vivo culture, 25% of the cells were prepared for flow cytometric analysis and 75% of the cells were prepared for competitive repopulation analysis from each well. Flow cytometric analysis and competitive repopulation analysis were performed as described above.

## Immunohistochemical analysis

Bone marrow sections were prepared as described previously[37]. In brief, bone samples were fixed in 4% paraformaldehyde, decalcified in DCAL medium, embedded in CPT medium, and frozen in cooled hexane. Alternatively, sections of undecalcified femoral bone were generated by Kawamoto's film method[38] (Cryofilm transfer kit; Section-Lab). Twelve-micrometer sections of femoral bone were generated using a CM3050S cryostat (Leica). For imaging of hematopoietic progenitors from Evi1-GFP mice, CD150⁺CD48⁻LSK HSCs, MPPs, GMPs, and MEPs were isolated from femurs and tibiae and fixed in 2% paraformaldehyde. Confocal microscopy was performed with an LSM780 (Zeiss). Confocal tiled images were analyzed using Bitplane Imaris software. Random spots were generated as described before[39]. In brief, random spots were generated from speckle noise detected in confocal images taken by the photomultiplier whose voltage was set to a value >900 with no sample set in the sample holder. The nuclei of cells were labeled with DAPI dye. Antibodies used for the immunohistochemical analysis are listed in Supplementary Table 2 in Supplementary Information file.

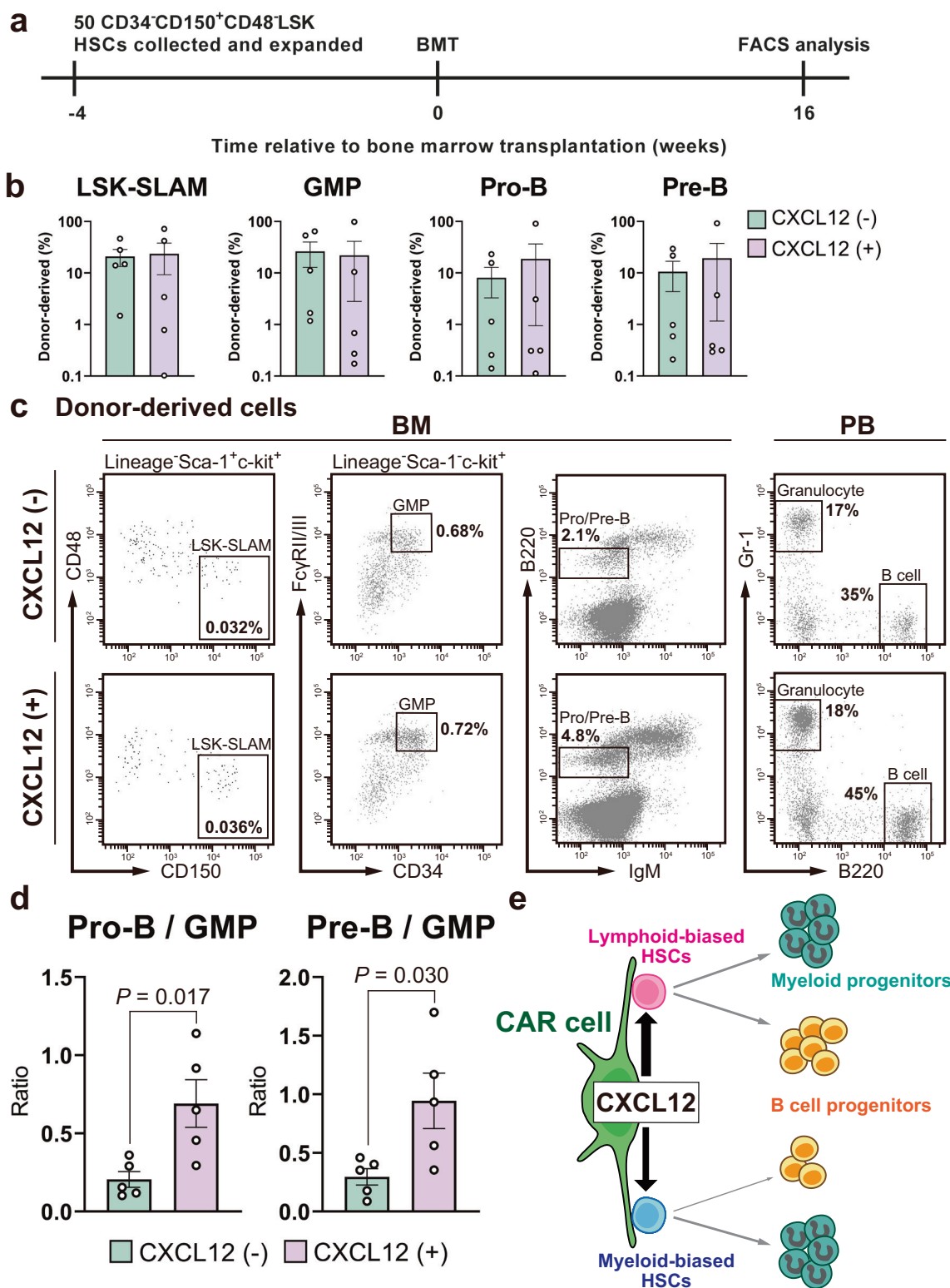

### RT-PCR analysis

Relative mRNA expression was analyzed by qRT-PCR analysis performed with a Step One Plus (Applied Biosystems) using Thunderbird SYBR qPCR mix (Toyobo). Total RNA was isolated from cells using Isogen (Nippon Gene) and treated with DNase I (Invitrogen), and cDNA was synthesized using Super Script VILO (Invitrogen) following the manufacturer's instructions. Values for each gene were normalized to the relative quantity of GAPDH mRNA in each sample. The primers used for qRT-PCR are listed in Supplementary Table 3 in Supplementary Information file.

### Statistical analysis

For comparisons between two groups, unpaired two-tailed Student's *t* test was used. Statistical significances were calculated using GraphPad

**Fig. 5 | CXCL12 enabled the maintenance of B lineage repopulating ability of HSCs in vitro. a** Experimental design. Sorted 50 cells in the HSC population from CD45.2⁺ wild-type mice were cultured for 28 days in the presence of SCF and TPO with or without CXCL12, and 75% of cells in each well were transplanted with 1×10⁶ CD45.1⁺ competitor bone marrow cells into CD45.1⁺CD45.2⁺ wild-type mice. **b** Donor chimerism of LSK-SLAM HSCs, GMPs, pro-B cells, and pre-B cells in the bone marrow of recipients at 16 weeks after transplantation is shown (*n* = 5). All error bars represent SE of the mean. **c** Representative flow cytometry plots of bone

marrow and peripheral blood from recipient mice at 16 weeks after transplantation. **d** Donor-derived B lymphoid/myeloid (pro-B/GMP and pre-B/GMP) ratios in recipients at 16 weeks after transplantation are shown (*n* = 5). All error bars represent SE of the mean. Statistical significances were calculated using the two-tailed unpaired Student's *t* test. Source data are provided as a Source data file. **e** Working model for how CXCL12 regulates HSCs to produce the required number of B cell progenitors. CAR cell-derived CXCL12 plays a critical role in the maintenance of HSCs, especially lymphoid-biased HSCs.

Prism 9.3.1 (GraphPad Software). All experiments were repeated at least three times with sufficient reproducibility.

## Reporting summary
Further information on research design is available in the Nature Portfolio Reporting Summary linked to this article.

## Data availability
Source data are provided with this paper.

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

## Acknowledgements
We thank N. Fujii and S. Oishi for the gift of chemically synthesized CXCL12, K. Nagahara for secretarial assistance, and members of the Center for Animal Experiments, Institute for Life and Medical Sciences, Kyoto University for animal care. This work was supported by grants from the JSPS KAKENHI (grant number 18H03998 and 17H05643 to T. Nagasawa, 19K08837 to T.S., and 22H05064 and 22H02850 to Y.O.), Mitsubishi Foundation award to T. Nagasawa, and Cooperative Research Program (Joint Usage/Research Center program) of Institute for Frontier Life and Medical Sciences, Kyoto University to T. Nagasawa.

## Author contributions
T. Nakatani, T.S., Y.O., and T. Nagasawa designed the experiments. T. Nakatani, T.S., and T. Nagasawa performed the experiments, analyzed the data, and prepared the paper. H.W. and G.K. contributed materials and tools. T. Nagasawa supervised the study. All authors discussed results and edited the manuscript.

## Competing interests
The authors declare no competing interests.
