## [Peer Review File · Nature Communications]

REVIEWER COMMENTS

Reviewer #1 (Remarks to the Author):

The authors used an inducible model of Cxcl12 deletion from LepR+/CAR cells, which form the niche for hematopoietic stem cells (HSCs) and early restricted progenitors in the bone marrow. The authors observed very little HSC depletion in young adult mice but profound depletion of HSCs from old mice. They observed a modestly increased depletion of B lineage progenitors than myeloid progenitors. The authors suggested that Cxcl12 from CAR cells was more important for the maintenance of lymphoid-biased as compared to myeloid-biased HSCs, and that addition of Cxcl12 to ex vivo cultures increased lymphoid reconstituting capacity after competitive transplantation of the cells.

1. This paper has multiple compelling strengths. While we reported limited HSC depletion from the bone marrow of young adult mice after Cxcl12 deletion from LepR+ cells, Nakatani et al. show a much more profound depletion of HSCs and restricted hematopoietic progenitors when these mice are aged for nearly 2 years. In my opinion, this is the most interesting and important observation in the paper because it shows that Cxcl12 from LepR+/CAR cells is even more important for the maintenance of HSCs than could be appreciated from prior studies. This result by itself is worthy of publication in Nature Communications.

2. Another strength of the paper is that the authors created a new reporter allele that distinguishes between the cells in which Cxcl12 was deleted versus the cells in which it wasn't. It is rare for studies to put this level of effort into generating a new allele that indicates gene deletion.

3. The title seems over-interpreted as it suggests that Cxcl12 is selectively required for the maintenance of lymphoid biased HSCs. The authors observed widespread depletion of myeloid lineage progenitors in addition to lymphoid progenitors. So it's required for both, though the phenotype appears to be a bit stronger among lymphoid progenitors. The title shouldn't imply that it's only important for lymphoid lineage progenitors.

4. The authors concluded that Cxcl12 was more important for lymphoid biased HSCs than for myeloid biased HSCs. This is mainly based on transplantation experiments where 200 sorted HSCs from Cxcl12 mutant mice were transplanted competitively into irradiated recipient mice. Bone marrow and peripheral blood chimerism demonstrated myeloid biased reconstitution by HSCs derived from the Cxcl12 mutant donors. However, the donor mice also exhibited a depletion of myeloid lineage cells in the bone marrow. These results are interesting but there are alternative interpretations. For example, lymphoid biased HSCs have undergone more rounds of division than myeloid biased HSCs (myeloid biased HSCs are the more primitive and quiescent HSCs that give rise to lymphoid biased HSCs as HSC self-renewal potential is eroded by cell division). Could the depletion of lymphoid biased HSCs and lymphoid progenitors from the Cxcl12 mutant mice reflect increased cell division by these cells in an effort to repair the hematopoietic defects in these mice? If so, this would reflect an indirect effect of Cxcl12 depletion rather than a direct effect of the niche on lymphoid lineage progenitors. Thus, it's not clear what this observation means and whether it reflects an indirect or direct effect. Currently, the authors build the manuscript around these data. In my view, the manuscript would be improved if the

authors could soften these conclusions and avoid suggesting that Cxcl12 mainly regulates lymphoid-lineage cells.

5. The authors suggest that HSCs detached from CXCL12^{-/-} LepR⁺/CAR cells and migrated to CXCL12-intact LepR⁺/CAR cells in the bone marrow. However, the data are indirect. There is no evidence of actual HSC migration and the authors did not show data indicating that HSCs are depleted within 5um of Cxcl12-intact LepR⁺/CAR cells. The authors should either provide data demonstrating that HSCs are depleted near CXCL12^{-/-} CAR cells or eliminate this conclusion.

6. The concern about the strength of the data related to HSC localization is amplified by uncertainty about the specificity of the Evi1 reporter. CD150⁺CD48⁻ cells are impressively enriched among the Evi1 reporter positive cells, but the authors didn't test what fraction of HSCs are reporter positive in transplantation assays or what percentage of the Evi1⁺c-kit⁺ cells are HSCs by competitively transplanting small numbers of these cells (e.g. 1-3 cells per recipient). Evi1 is expressed by other primitive hematopoietic progenitors (including MPPs, GMPs, and MEPs) at least at the RNA level, and these cells also express c-kit.

Reviewer #2 (Remarks to the Author):

In this study, Nakatany et al., investigate the role of CAR cells derived Cxcl12 in middle-aged and aged mice and propose that Cxcl12 selectively regulates lymphoid-biased HSCs. The authors generated a series of new transgenic mouse models to conditionally delete Cxcl12, Cre lines to achieve recombination in CAR cells and an Evi1-GFP reporter to identify HSCs in situ. The most interesting findings in this study are related to the lymphoid skewing of HSCs isolated from Cxcl12 deficient mice. However, the authors only provide a very superficial analysis which does not definitively demonstrate that this skewing results from a selective role of Cxcl12 in true lymphoid-biased HSCs. The new transgenic mouse models require further validation that they are indeed functioning as suggested. The choice of analysing mice 14-23 weeks post Tamoxifen administration is a confounding factor as these mice are no longer young and not old either. More specific comments below.

1. HSCs isolated from Cxcl12 deficient mice have overall very low multilineage reconstitution efficiency across all lineages and these findings are highly compatible with bone marrow failure resulting from no HSC reconstitution. This is especially important given the very high number of LSK-SLAM HSCs transplanted (200 cells per mouse). The authors need to perform secondary transplantations to ensure the observed reconstitution in the primary recipients indeed represents HSC activity.

2. The authors conclude that Cxcl12 deficiency is affecting lymphoid biased HSCs because the B-cell lineage was more affected. However, the existence of lymphoid-biased HSCs has been recently challenged as no lymphoid-biased patterns of reconstitution have been observed from single HSCs (PMID: 29298288). What is happening to the different subsets of lineage-biased MPPs (MPP1-4)?

3. In Fig 5 the authors show that HSCs cultured in vitro in the absence of Cxcl12 are affected in their capacity to reconstitute the lymphoid lineage but only a ratio of PreB/GMP is shown. Please show full analysis of HSCs BM reconstitution after 16 weeks and peripheral blood analysis of B and myeloid reconstitution, to demonstrate that long-term HSC reconstitution was efficient.

4. The authors use S100 expression to define CAR cells. Full phenotypic definition of CAR cells used for their analysis/isolation by Flow cytometry and in imaging analyses needs to be shown. How specific is S100 expression for CAR cells? And what is the overlap between S100 and other standard CAR cell markers like *Lepr*, Cxcl12-GFP and the new Cxcl12-TdTomato model described here?

5. Contrarily to all other Cxcl12 deletion models in this study, the Cxcl12-Tomato-FL x Ebf3-Cre mice in Fig.1f do not show reduction in HSC numbers. There is however a very high variation between the only 3 biological replicates analysed. This level of variation does not allow the conclusion that there is no difference and more biological replicates need to be added to this figure. Furthermore, the fold-reduction of TdTomato cells (Fig.1d) is much lower than the fold-reduction in Cxcl12 mRNA observed in Fig.1a. The authors need to validate that tomato expression indeed fully correlates with Cxcl12 mRNA abundance, because if the half-life of TdTomato is higher than the half-life of Cxcl12-mRNA there might be Tomato+ cells that have been deleted.

6. This study describes a new HSC reporter mouse model. The characterization of these mice should be extended. The authors show efficient reconstitution out of LSK-SLAM GFP+ cells (although this was low given that 200 cells have been transplanted) However, the definition used for imaging is Evi1-GFP+ Kit+ so this definition should be used to validate efficiency reconstitution potential, an ideally in a limiting dilution and competitive assay, to fully quantify HSC activity in this population. Since this is a knock-in model, is Evi1 deficiency affecting HSC function? In Ext Data Fig.2d please present full LSK-SLAM gating strategy. What haematopoietic cell types are present in the remaining GFP+ cells? What is the LSK-SLAM purity if Kit+ GFP low cells are also included in the gate?

7. To investigate the localization of HSCs near CAR cells the authors transplant Evi1-GFP HSCs into Cxcl12-Tomato:Ebf3-Cre mice. However, irradiation has a long-term effect on the perivascular niche and HSCs will be activated and highly proliferative upon transplantation. These experiments should be performed by crossing the Evi1-GFP reporter mice with the Cxcl12-Tomato-FL/EBF3-Cre mice to investigate the localization of HSCs in their native niche.

8. The authors conclude that HSCs have moved closer to CAR cells that preserved intact expression of Cxcl12 but this was only analysed indirectly in Fig1h by measuring numbers of HSCs near intact CAR cells. The number of HSC near CAR cells that lost Cxcl12 expression needs to be analysed to definitively conclude HSCs have moved away. How many HSCs per mouse have been analysed in these

experiments? Full analysis of distances of HSCs to CAR cells should be performed to present the shift in the average difference of HSCs to CAR cells that lost Cxcl12 expression.

9. What is the rationale used for choosing the age of mice, especially the middle age analysis? Why were the aged mice analysed with Lepr-Cre and not with the same Ebf3-CreERT2?

10. Why do the authors wait 17-23 weeks to analyse the mice following tamoxifen administration? Is the niche remodelled after >17 weeks post Tamoxifen, given that there are so many defects in haematopoiesis? Is the endosteal niche, that has previously been shown to support early lymphoid cells affected and in this way indirectly affect the lymphoid output?

Minor:

- The authors should show representative examples of all flow cytometric gating strategies. FACS analysis shown in Extended data Fig2b has serious compensation issues (at least on the lineage channel) and this should be corrected.

- Please extend the characterization of the Evi1-GFP mouse model in Ext. Data Fig 1. This should

- More examples of images of Evi1-GFP HSC in relation to CAR cells should be added to supplements.

- CD34 is used to define HSCs. However CD34 expression in HSCs changes with cell cycle status and Cxcl12 deficiency interferes with the quiescence of HSCs and induces their mobilization. Analysis should be performed excluding this marker.

Reviewer #3 (Remarks to the Author):

The manuscript by Nakatani et al suggests that CXCL12 selectively regulates lymphoid-biased hematopoietic stem cells (HSCs). The authors use a variety of methods including different Cre lines to eliminate CXCL12 in niche cell populations, long-term repopulations assays, cell culture and immunofluorescence studies. The paper is interesting because it includes variable age and long follow up. However, most experiments have been performed without separation of HSCs (myeloid-biased or balanced/lymphoid biased) and a number of concerns limit the extent of the conclusions that can be obtained with the data available, as explained below. As a result the main conclusions do not seem fully supported and the overall knowledge appears incremental over the current literature.

Major comments:

1) Introduction, L42-44. The first reported evidence that mouse MSCs are important in the HSC niche is PMID: 20703299 (not cited).

2) Evi1-GFP alone cannot be used as an HSC marker. FigS2d shows intermediate GFP expression (which cannot be distinguished from GFP^{hi} by immunofluorescence) in cells that are not HSCs, which is consistent with the original report showing Evi-GFP expression in LSK cells, where only a small proportion would be HSCs (PMID: 22084405). The conclusions of Figure 1 are pertinent to progenitors (HSPCs, LSK cells), but cannot be extrapolated to HSCs in the absence of additional markers. In fact, the authors' experiments show that Evi1 mRNA expression is not reduced in the BM of mice with extensive HSC depletion (Fig. 3c-d), further emphasizing that Evi1 expression alone does not reflect HSCs. What's the number of cells imaged in Fig 1H? In the Ebf3-CreERT2 data in Figure 1b, what type of cells increase in the KO, since the total number is similar, yet some reduction was indicated in HSCs and B cell progenitors? Fig2b/3b- Given the overall cellularity is decreased, does the frequency of the populations also change to see if there is a differentiation bias, rather than by cell number alone? Were any differences in blood counts observed?

3) I believe that additional data is required to interpret that CXCL12 from BM LepR⁺/CAR cells is essential for HSC maintenance based on the results of Fig.2. 1st, LepR⁺ cells are also present in other organs (spleen, PMID: 26570997, etc) where they produce CXCL12. It is unclear how much locally-produced vs. plasma levels of CXCL12 are affected in young/old mice to be able to interpret the results and support the authors' conclusion. 2nd, aged mice have massively reduced endosteal niches (PMID: 31303548; PMID: 31685996) but this does not seem to be accounted for and could affect the overall degree of recombination in LepR⁺ cells and also explains the larger CXCL12 depletion, compared with young mice. The larger CXCL12 depletion, the less HSCs, as elegantly demonstrated by this group in the original Nature paper; but this does not demonstrate that these LepR⁺/CAR cells are essential to maintain HSCs through CXCL12 in any way in the young mouse. Furthermore, which cells are the authors referring to? Exclusiv perisinusoidal cells as initially claimed (REF 7)? Or periarteriolar cells also marked by LepR by the same group in a later publication (PMID: 33627868)? MSCs or other bone-forming cells derived from MSCs, such as preosteoblasts, osteoblasts, osteocytes ? – all of them are targeted by LepR-Cre and therefore it is difficult to know which cell is doing what or extrapolate conclusions from overall deletion in all these different cell types producing CXCL12.

4) To compare the reconstitution potential of myeloid-biased or balanced(“=lymphoid-biased”) HSCs the authors should have sorted each HSC type based on the markers available and compare their function. It is very difficult to infer which HSC population has contributed to hematopoietic reconstitution after transplanting the unseparated, bulk HSCs. The frequency of CD150^{low} or CD229^{high} cells is less reduced than the frequency of overall LT-HSCs in Fig.4, so the claim that lymphoid-bias HSCs are especially affected does not seem supported. Multiple myeloid lineages were profoundly depleted as well, which does not support the overall conclusion.

5) Fig4- Do they have the chimerism data throughout the transplant- Wk 4/8/12 to see if chimerism was always low or if it just dipped at the end? They also only show select subsets of cells, is there a reason why- e.g. why Pre B over other B cell populations, could they include an overall summary in the supplementary? To support ratio data, are the frequency of the populations altered in donor cells to confirm differentiation bias. Figure 4 lacks a depth of how the naive mature B cells look like in the secondary lymphoid organs such as spleen and lymph nodes. And not just reduced numbers, do they have a defect on function(i.e., antibody production after challenging with antigen, or response to LPS or anti-CD40).

6) Fig5- Following the culture- was there a difference in cell number at the end of the culture- how many cells were transplanted- was apoptosis rate different with/without CXCL12? Did the authors carry out FACS analysis at the end of the culture to see what cells were being transplanted? To investigate if CXCL12 affected cells differentiation and how many HSCs and other progenitors were transplanted? Was there any more FACS analysis carried out at the end of the 16-week transplant? Blood counts? Frequency of subsets?

Minor comments:

- L.70. Delete “the former mutants”
- Lines 69-71: Same mouse model stated ‘Ebf3-CreERT2;CXCL12f/f mice’, should one be f/-?
- Figure axis throughout could be clearer/ more detailed e.g ‘Cxcl12 mRNA (fold)’ or ‘Frequency of LT-HSCs’.
- Supplementary Fig2c- Could the authors show the negative GFP mouse or unstained control for the gating?
- Line 121- Please briefly define ‘random spots near intact CAR cells’ in text or methods, methods reference a previous paper.
- Fig1f- Is frequency also unaltered or total cell number for context?
- When there are numerous graphs per letter in the figure, should be broken up- e.g Fig1b, Fig 2b etc

Reviewer #1 (Remarks to the Author):

We would like to express our thanks to the reviewer for reviewing this manuscript and for the helpful comments.

The authors used an inducible model of Cxcl12 deletion from LepR⁺/CAR cells, which form the niche for hematopoietic stem cells (HSCs) and early restricted progenitors in the bone marrow. The authors observed very little HSC depletion in young adult mice but profound depletion of HSCs from old mice. They observed a modestly increased depletion of B lineage progenitors than myeloid progenitors. The authors suggested that Cxcl12 from CAR cells was more important for the maintenance of lymphoid-biased as compared to myeloid-biased HSCs, and that addition of Cxcl12 to ex vivo cultures increased lymphoid reconstituting capacity after competitive transplantation of the cells.

1. This paper has multiple compelling strengths. While we reported limited HSC depletion from the bone marrow of young adult mice after Cxcl12 deletion from LepR⁺ cells, Nakatani et al. show a much more profound depletion of HSCs and restricted hematopoietic progenitors when these mice are aged for nearly 2 years. In my opinion, this is the most interesting and important observation in the paper because it shows that Cxcl12 from LepR⁺/CAR cells is even more important for the maintenance of HSCs than could be appreciated from prior studies. This result by itself is worthy of publication in Nature Communications.

We would thank again to the reviewer for the positive comments.

2. Another strength of the paper is that the authors created a new reporter allele that distinguishes between the cells in which Cxcl12 was deleted versus the cells in which it

wasn't. It is rare for studies to put this level of effort into generating a new allele that indicates gene deletion.

We would thank again to the reviewer for the positive comments.

3. The title seems over-interpreted as it suggests that Cxcl12 is selectively required for the maintenance of lymphoid biased HSCs. The authors observed widespread depletion of myeloid lineage progenitors in addition to lymphoid progenitors. So it's required for both, though the phenotype appears to be a bit stronger among lymphoid progenitors. The title shouldn't imply that it's only important for lymphoid lineage progenitors.

We agree. To address this, we replaced the word “selectively” with the word “preferentially” in the title.

4. The authors concluded that Cxcl12 was more important for lymphoid biased HSCs than for myeloid biased HSCs. This is mainly based on transplantation experiments where 200 sorted HSCs from Cxcl12 mutant mice were transplanted competitively into irradiated recipient mice. Bone marrow and peripheral blood chimerism demonstrated myeloid biased reconstitution by HSCs derived from the Cxcl12 mutant donors. However, the donor mice also exhibited a depletion of myeloid lineage cells in the bone marrow. These results are interesting but there are alternative interpretations. For example, lymphoid biased HSCs have undergone more rounds of division than myeloid biased HSCs (myeloid biased HSCs are the more primitive and quiescent HSCs that give rise to lymphoid biased HSCs as HSC self-renewal potential is eroded by cell division). Could the depletion of lymphoid biased HSCs and lymphoid progenitors from

the Cxcl12 mutant mice reflect increased cell division by these cells in an effort to repair the hematopoietic defects in these mice? If so, this would reflect an indirect effect of Cxcl12 depletion rather than a direct effect of the niche on lymphoid lineage progenitors. Thus, it's not clear what this observation means and whether it reflects an indirect or direct effect. Currently, the authors build the manuscript around these data. In my view, the manuscript would be improved if the authors could soften these conclusions and avoid suggesting that Cxcl12 mainly regulates lymphoid-lineage cells.

We would thank again to the reviewer for the helpful comments. We agree that our results exclude the possibility that continued reduction of HSCs increased cell division of lymphoid-biased HSCs in an effort to repair the hematopoietic defects, resulting in reduced generation of B cell progenitors from remaining lymphoid-biased HSCs. Thus, we stated this (**Page 14, lines 311 to 314**), softened our conclusions and avoided suggesting that Cxcl12 mainly regulates lymphoid-lineage cells in the abstract (**Page 2, line 27 and lines 31 to 32**) and text (**Page 4, line 80; Page 11, line 255; Page 14, line 321**).

5. The authors suggest that HSCs detached from CXCL12^{-/-} LepR⁺/CAR cells and migrated to CXCL12-intact LepR⁺/CAR cells in the bone marrow. However, the data are indirect. There is no evidence of actual HSC migration and the authors did not show data indicating that HSCs are depleted within 5um of Cxcl12-intact LepR⁺/CAR cells. The authors should either provide data demonstrating that HSCs are depleted near CXCL12^{-/-} CAR cells or eliminate this conclusion.

As requested, we added the data demonstrating that HSCs are depleted near CXCL12^{-/-} CAR cells in the text (**Page 7, lines 141 to 149**) and **Figure 1k, and 1l**.

6. The concern about the strength of the data related to HSC localization is amplified by uncertainty about the specificity of the Evi1 reporter. CD150⁺CD48⁻ cells are impressively enriched among the Evi1 reporter positive cells, but the authors didn't test what fraction of HSCs are reporter positive in transplantation assays or what percentage of the Evi1⁺c-kit⁺ cells are HSCs by competitively transplanting small numbers of these cells (e.g. 1-3 cells per recipient). Evi1 is expressed by other primitive hematopoietic progenitors (including MPPs, GMPs, and MEPs) at least at the RNA level, and these cells also express c-kit.

As requested, we tested what fraction of HSCs are reporter positive in transplantation assays or what percentage of the Evi1-GFP^{hi}c-kit⁺ cells are HSCs by measuring the competitive repopulating units (CRUs) based on limiting-dilution of competitive repopulation assays.

a. Frequency of CRU in the Evi1-GFP^{hi}c-kit⁺ cells was 1 in 3.8 (95% CI).

We stated this in **Supplementary Fig. 2e**.

b. Flow cytometric analysis revealed that CD150⁺CD48⁻ LSK cells (LT-HSCs) expressed high levels of Evi1-GFP but MPPs expressed lower levels of Evi1-GFP compared to LT-HSCs (**Supplementary Fig. 2b**) and that other primitive hematopoietic progenitors, including GMPs and MEPs, did not express Evi1-GFP (**Supplementary Fig. 2b**). We could detect the fluorescence signals in fixed LT-HSCs but not in fixed MPPs isolated from Evi1-GFP mice by histology (**Supplementary Fig. 3**).

Reviewer #2 (Remarks to the Author):

We would like to express our thanks to the reviewer for reviewing this manuscript and for the helpful comments.

In this study, Nakatany et al., investigate the role of CAR cells derived Cxcl12 in middle-aged and aged mice and propose that Cxcl12 selectively regulates lymphoid-biased HSCs. The authors generated a series of new transgenic mouse models to conditionally delete Cxcl12, Cre lines to achieve recombination in CAR cells and an Evi1-GFP reporter to identify HSCs in situ. The most interesting findings in this study are related to the lymphoid skewing of HSCs isolated from Cxcl12 deficient mice. However, the authors only provide a very superficial analysis which does not definitively demonstrate that this skewing results from a selective role of Cxcl12 in true lymphoid-biased HSCs. The new transgenic mouse models require further validation that they are indeed functioning as suggested. The choice of analysing mice 14-23 weeks post Tamoxifen administration is a confounding factor as these mice are no longer young and not old either. More specific comments below.

1. HSCs isolated from Cxcl12 deficient mice have overall very low multilineage reconstitution efficiency across all lineages and these findings are highly compatible with bone marrow failure resulting from no HSC reconstitution. This is especially important given the very high number of LSK-SLAM HSCs transplanted (200 cells per mouse). The authors need to perform secondary transplantations to ensure the observed reconstitution in the primary recipients indeed represents HSC activity.

As requested, we performed secondary transplantations. We transplanted 1500 donor-derived Lin⁻Sca-1⁻c-kit⁺ primitive hematopoietic stem and progenitor cells from primary recipients transplanted with 200 LSK-SLAM HSCs. HSCs from CXCL12^{ΔCAR} mice were present but severely reduced and their ability to generate B cell progenitors was markedly reduced after transplantation in secondary recipients.

We stated these in the text (**Page 9, lines 202 to 207; Page 11, lines 246 to 251**) and **Supplementary Fig. 9 and 11**.

2. The authors conclude that Cxcl12 deficiency is affecting lymphoid biased HSCs because the B-cell lineage was more affected. However, the existence of lymphoid-biased HSCs has been recently challenged as no lymphoid-biased patterns of reconstitution have been observed from single HSCs (PMID: 29298288). What is happening to the different subsets of lineage-biased MPPs (MPP1-4)?

a. As pointed out, it has been reported that lymphoid-biased HSCs are identical to and referred as to balanced HSCs. We stated this and cited the paper (PMID: 29298288) in the text (**Page 4, line 78 and 79**).

b. As requested, we showed frequencies and numbers of MPP1-4 in the bone marrow of tamoxifen-treated control and Ebf3-CreERT2;CXCL12^{f/-} (CXCL12^{ΔCAR}) mice.

Consistent with our conclusions, the numbers of MPP4s with high lymphoid and low megakaryocyte/erythroid potential were reduced; however, the numbers of MPP3s with high myeloid potential were unaltered and the numbers of MPP2s with low lymphoid and high megakaryocyte/erythroid potential were modestly increased. These results support our conclusions. We stated these in **Fig. 3d and Supplementary Fig. 8b**.

3. In Fig 5 the authors show that HSCs cultured in vitro in the absence of *Cxcl12* are affected in their capacity to reconstitute the lymphoid lineage but only a ratio of PreB/GMP is shown. Please show full analysis of HSCs BM reconstitution after 16 weeks and peripheral blood analysis of B and myeloid reconstitution, to demonstrate that long-term HSC reconstitution was efficient.

As requested, we showed long-term BM HSC reconstitution and peripheral blood B and myeloid reconstitution of HSCs cultured in vitro with and without CXCL12 at 16 weeks after transplantation. Percentages of donor-derived hematopoietic cells were variable between recipients but the donor B cell progenitor/myeloid progenitor reconstitution ratios in recipients of cultured cells with CXCL12 was significantly higher than those in recipients of cultured cells without CXCL12. We stated this in the text (**Page 12, lines 259 to 264**) and **Fig. 5b and 5c**.

4. The authors use *S100* expression to define CAR cells. Full phenotypic definition of CAR cells used for their analysis/isolation by Flow cytometry and in imaging analyses needs to be shown. How specific is *S100* expression for CAR cells? And what is the overlap between *S100* and other standard CAR cell markers like *Lepr*, *Cxcl12-GFP* and the new *Cxcl12-TdTomato* model described here?

a. We defined CAR cells as $\text{PDGFR}\beta^+\text{Sca-1}^-\text{CD31}^-\text{CD45}^-\text{Ter119}^-$ cells (Omatsu et al., *Immunity* 33;381,2010; Omatsu et al., *Nature* 508; 536, 2014) or *CXCL12-tdTomato*⁺ cells in the bone marrow by flow cytometry. In imaging analysis, we defined CAR cells

as S100⁺ cells, PDGFRβ⁺ cells (Omatsu et al., Nature 508; 536, 2014), or CXCL12-tdTomato⁺ cells in the bone marrow.

We stated the definition of CAR cells (**Page 3, lines 47 to 50**) and showed full phenotypic definition of CAR cells used for our analysis/isolation by flow cytometry (**Page 5, line 92; Page 8, line 161, line 183**) and in imaging analyses (**Page 7, lines 146 to 148**).

b. About 96% of S100⁺ cells were CXCL12-GFP⁺ CAR cells. Conversely, about 97% of CXCL12-GFP⁺ CAR cells were S100⁺ cells. In addition, using CXCL12-tdTomato knockin mice we showed that CXCL12-tdTomato⁺ CAR cells overlap strongly with S100⁺ cells in the bone marrow cavity. About 90% of S100⁺ cells were CXCL12-tdTomato⁺ CAR cells. Conversely, about 97% of CXCL12-tdTomato⁺ CAR cells were S100⁺ cells. We showed these in **Fig. 1 for review purposes**.

5. Contrarily to all other Cxcl12 deletion models in this study, the Cxcl12-Tomato-FL x Ebf3-Cre mice in Fig.1f do not show reduction in HSC numbers. There is however a very high variation between the only 3 biological replicates analysed. This level of variation does not allow the conclusion that there is no difference and more biological replicates need to be added to this figure. Furthermore, the fold-reduction of TdTomato cells (Fig.1d) is much lower than the fold-reduction in Cxcl12 mRNA observed in Fig.1a. The authors need to validate that tomato expression indeed fully correlates with Cxcl12 mRNA abundance, because if the half-life of TdTomato is higher than the half-life of Cxcl12-mRNA there might be Tomato⁺ cells that have been deleted.

a. As requested, we added more biological replicates in HSC numbers in Ebf3-CreERT2; Cxcl12-tdTomato^{ff} mice, indicating that HSC numbers were unaltered. We stated this in **Fig. 1f**.

b. The data in Fig. 1d are from Ebf3-CreERT2; Cxcl12-tdTomato^{ff} mice although the data in Fig. 1a are from Ebf3-CreERT2;Cxcl12^{f/-} mice. This could explain why the fold-reduction of tdTomato⁺ cell frequencies in Fig.1d is much lower than the fold-reduction in Cxcl12 mRNA levels observed in Fig.1a. We showed tdTomato expression fully correlates with CXCL12 mRNA abundance and thus the fold-reduction of tdTomato⁺ cell frequencies (Fig.1f) was similar to the fold-reduction in Cxcl12 mRNA levels observed in Ebf3-CreERT2; Cxcl12-tdTomato^{ff} mice (**Fig. 2 for review purposes**).

The fold-reduction of tdTomato⁺ cell frequencies and Cxcl12 mRNA levels in Ebf3-CreERT2; Cxcl12-tdTomato^{ff} mice was lower than that in Ebf3-CreERT2;Cxcl12^{f/-} mice. This may be because of differences in length of deleted genomic DNA between Ebf3-CreERT2; Cxcl12-tdTomato^{ff} mice and Ebf3-CreERT2;Cxcl12^{f/-} mice.

6. This study describes a new HSC reporter mouse model. The characterization of these mice should be extended. The authors show efficient reconstitution out of LSK-SLAM GFP⁺ cells (although this was low given that 200 cells have been transplanted) However, the definition used for imaging is Evi1-GFP⁺ Kit⁺ so this definition should be used to validate efficiency reconstitution potential, an ideally in a limiting dilution and competitive assay, to fully quantify HSC activity in this population. Since this is a knock-in model, is Evi1 deficiency affecting HSC function? In Ext Data Fig.2d please

present full LSK-SLAM gating strategy. What haematopoietic cell types are present in the remaining GFP⁺ cells? What is the LSK-SLAM purity if Kit⁺ GFP low cells are also included in the gate?

a. As requested, we quantified HSC activity in the Evi1-GFP^{hi} c-Kit⁺ population by measuring the competitive repopulating units (CRUs) based on limiting-dilution of competitive repopulation assays. Frequency of CRU in the Evi1-GFP^{hi}c-kit⁺ cells was 1 in 3.8 (95% CI). We stated this in the text (**Page 6, lines 109 to 111**) and

Supplementary Fig. 2e.

b. Since we generated Evi1-GFP knockin mice using a targeting vector bearing a internal ribosomal entry site (IRES) that flanked exon 15 of the Evi1 gene, expression of endogenous Evi1 was retained and HSC numbers were unaltered in the reporter mice. We stated this in the text (**Page 6, lines 119 to 120**) and **Supplementary Fig. 4.**

c. As requested, we show full LSK-SLAM gating strategy in previous Ext Data Fig.2d in **Supplementary Fig. 2d.**

d. Flow cytometric analysis revealed that 84% of cells expressing high levels of Evi1-GFP were LSK CD150⁺CD48⁻ LT-HSCs and others were LSK CD150⁺CD48⁻ multipotent progenitors (MPPs). However, most MPPs expressed lower levels of Evi1-GFP compared to LT-HSCs (**Supplementary Fig. 2b**). Other primitive hematopoietic progenitors, including GMPs and MEPs, did not express Evi1-GFP (**Supplementary Fig. 2b**).

81 % of Evi1-GFP⁺c-kit⁺ cells, including Evi1-GFP^{lo}c-kit⁺ cells were LSK CD150⁺CD48⁻ cells.

However, importantly, we could detect the fluorescence signals in fixed LT-HSCs but not in fixed MPPs isolated from Evi1-GFP mice by histology (**Supplementary Fig. 3**), indicating that Evi1-GFP mice allow visualization of HSCs in the bone marrow sections.

7. To investigate the localization of HSCs near CAR cells the authors transplant Evi1-GFP HSCs into Cxcl12-Tomato:Ebf3-Cre mice. However, irradiation has a long-term effect on the perivascular niche and HSCs will be activated and highly proliferative upon transplantation. These experiments should be performed by crossing the Evi1-GFP reporter mice with the Cxcl12-Tomato-Fl/EBF3-Cre mice to investigate the localization of HSCs in their native niche.

Since some endothelial cells expressed GFP in the Evi1-GFP HSC-reporter mice, we transplant hematopoietic cells, including HSCs from the Evi1-GFP HSC-reporter mice into irradiated Ebf3-CreERT2;CXCL12-tdTomato f/f mice to investigate the localization of HSCs and CAR cells. Irradiation might have a long-term effect on the perivascular niche and HSCs; however, we revealed that the perivascular niche and HSCs from irradiated mice were largely unaltered after 16 weeks following transplantation when we analyzed bone marrow. We showed that expression of HSC niche factors, including CXCL12 and SCF in CAR cells and endothelial cells, and transcription factors essential for HSC niche maintenance in CAR cells, CXCL12 and SCF, the frequencies of CAR cells, HSC numbers, and frequencies of cycling HSCs in

the bone marrow of irradiated mice were comparable with those of untreated mice in **Fig. 3 for review purposes.**

8. The authors conclude that HSCs have moved closer to CAR cells that preserved intact expression of Cxcl12 but this was only analysed indirectly in Fig1h by measuring intact CAR cells. The number of HSC near CAR cells that lost Cxcl12 expression needs to be analysed to definitively conclude HSCs have moved away. How many HSCs per mouse have been analysed in these experiments? Full analysis of distances of HSCs to CAR cells should be performed to present the shift in the average difference of HSCs to CAR cells that lost Cxcl12 expression.

a. As requested, we analyzed the frequencies of HSCs near CAR cells that lost Cxcl12 expression. Although only 42% of CAR cells were CXCL12-intact, 85 % of HSCs were located near Cxcl12-intact CAR cells and only 20 % of HSCs were located near PDGFR β ⁺tdTomato⁻ CAR cells that lost Cxcl12 expression. These results support our conclusions. We stated these in the text (**Page 7, lines 141 to 149**) and **Figure 1h, 1k, and 1l.**

b. We analyzed 960 and 1030 HSCs in control CXCL12-tdTomato^{ff} and Ebf3-CreERT2;CXCL12-tdTomato^{ff} mice, respectively, in these experiments.

c. As requested, we showed full analysis of distances of HSCs to CAR cells to present the shift in the average difference of HSCs to CAR cells that lost Cxcl12 expression. We stated these in **Figure 1g, 1j, and 1l.**

9. What is the rationale used for choosing the age of mice, especially the middle age analysis? Why were the aged mice analysed with Lepr-Cre and not with the same Ebf3-CreERT2?

The aim of this study is to determine the phenotypes of HSCs in adult animals whose CXCL12 is deleted from all CAR cells. Thus, we analyzed 20-24-week-old tamoxifen-treated Ebf3-CreERT2;CXCL12^{f/-} mice. In the analysis of LepR-Cre;CXCL12^{f/-} mice, we analyzed 90-week-old animals because CXCL12 was deleted from about 99.5% of the CAR cells in these mice although CXCL12 was deleted from only about 90% of the CAR cells in 20-24-week-old mutants. Similar results were obtained when we analyzed 90-week-old tamoxifen-treated Ebf3-CreERT2;CXCL12^{f/-} mice, in which CXCL12 is deleted from all CAR cells.

10. Why do the authors waited 17-23 weeks to analyse the mice following tamoxifen administration? Is the niche remodelled after >17 weeks post Tamoxifen, given that there are so many defects in haematopoiesis? Is the endosteal niche, that has previously been shown to support early lymphoid cells affected and in this way indirectly affect the lymphoid output?

We believe that 17-23 weeks are sufficient for bone marrow hematopoietic niches to restore and sufficient and required for short-term HSCs and hematopoietic progenitors generated in wild-type hematopoietic niches before tamoxifen treatment to disappear. 17-23 weeks after tamoxifen treatment, most HSCs and hematopoietic cells would be regulated by microenvironments where CXCL12 was deleted in almost all CAR cells.

It has been previously reported that the endosteal niche is involved in generation and/or maintenance of CLPs but not B cell precursors, including pro-B and pre-B cells (Ding and Morrison, Nature 495;231,2013).

Minor:

- The authors should show representative examples of all flow cytometric gating strategies. FACS analysis shown in Extended data Fig2b has serious compensation issues (at least on the lineage channel) and this should be corrected.

As requested, we corrected compensation issues on the lineage channel and showed representative examples of all flow cytometric gating strategies in **Supplementary Fig. 12**.

- Please extend the characterization of the Evi1-GFP mouse model in Ext. Data Fig 1. This should

As requested, we extend the characterization of the Evi1-GFP mouse model and showed additional data in **Supplementary Fig. 2 and 3**.

- More examples of images of Evi1-GFP HSC in relation to CAR cells should be added to supplements.

As requested, we added more examples of images of Evi1-GFP HSC in relation to CAR cells in **Supplementary Fig. 6**.

- CD34 is used to define HSCs. However CD34 expression in HSCs changes with cell cycle status and Cxcl12 deficiency interferes with the quiescence of HSCs and induces their mobilization. Analysis should be performed excluding this marker.

As requested, we added the data on HSCs performed excluding CD34 in the text (**Page 5, line 97**), **Fig. 2c, 3c, 4a, 5b, and Supplementary Fig. 4, 5, 7a, 8a, 10b, 10c, 11a, 12a, and 12b.**

Reviewer #3 (Remarks to the Author):

We would like to express our thanks to the reviewer for reviewing this manuscript and for the helpful comments.

The manuscript by Nakatani et al suggests that CXCL12 selectively regulates lymphoid-biased hematopoietic stem cells (HSCs). The authors use a variety of methods including different Cre lines to eliminate CXCL12 in niche cell populations, long-term repopulations assays, cell culture and immunofluorescence studies. The paper is interesting because it includes variable age and long follow up. However, most experiments have been performed without separation of HSCs (myeloid-biased or balanced/lymphoid biased) and a number of concerns limit the extent of the conclusions that can be obtained with the data available, as explained below. As a result the main conclusions do not seem fully supported and the overall knowledge appears incremental over the current literature.

Major comments:

1) Introduction, L42-44. The first reported evidence that mouse MSCs are important in the HSC niche is PMID: 20703299 (not cited).

We agree with the comment and stated this and cited the paper (PMID: 20703299) **(Page 3, lines 41 to 44)**.

2) Evi1-GFP alone cannot be used as an HSC marker. FigS2d shows intermediate GFP expression (which cannot be distinguished from GFPhi by immunofluorescence) in cells that are not HSCs, which is consistent with the original report showing Evi-GFP expression in LSK cells, where only a small proportion would be HSCs (PMID: 22084405). The conclusions of Figure 1 are pertinent to progenitors (HSPCs, LSK cells), but cannot be extrapolated to HSCs in the absence of additional markers. In fact, the authors' experiments show that Evi1 mRNA expression is not reduced in the BM of mice with extensive HSC depletion (Fig. 3c-d), further emphasizing that Evi1 expression alone does not reflect HSCs. What's the number of cells imaged in Fig 1H? In the Ebf3-CreERT2 data in Figure 1b, what type of cells increase in the KO, since the total number is similar, yet some reduction was indicated in HSCs and B cell progenitors? Fig2b/3b- Given the overall cellularity is decreased, does the frequency of the populations also change to see if there is a differentiation bias, rather than by cell number alone? Were any differences in blood counts observed?

a. Most cells in the LT-HSC population expressed high levels of Evi1-GFP (**Supplementary Fig. 2b**). However, as pointed out, most Lin⁻Sca-1⁻c-kit⁺CD150⁻CD48⁻ multipotent progenitors (MPPs) expressed lower levels of Evi1-GFP compared to LT-HSCs by flow cytometry (**Supplementary Fig. 2b**). Other primitive hematopoietic

progenitors, including GMPs and MEPs, did not express Evi1-GFP (**Supplementary Fig. 2b**). Importantly, we could detect the fluorescence signals in fixed LT-HSCs but not in fixed MPPs isolated from Evi1-GFP mice by histology (**Supplementary Fig. 3**). Thus, we can distinguish Evi1-GFP^{high}c-kit⁺ cells, which are enriched for HSCs (**Supplementary Fig. 2d and 3**), from Evi1-GFP intermediate or low cells, including MPPs in the bone marrow sections by immunofluorescence.

b. It is incorrect to say that our experiments showed that Evi1 mRNA expression was not reduced in the BM cells of mice with extensive HSC depletion because we show that Evi1 mRNA expression is not reduced in the sorted residual HSCs from mice with extensive HSC depletion. Our results are consistent with the fact that Evi1 is preferentially expressed in HSCs and indicate that CAR cell-derived CXCL12 did not affect expression of Evi1 in HSCs. Thus, Evi1-GFP mice allowed us to visualize remaining HSCs of mice lacking CXCL12 in all CAR cells.

c. The numbers of cells imaged for Fig. 1j (Fig. 1h in the original manuscript) are 960 (control) and 1030 (mutants).

d. Since HSC numbers are much smaller than those of total bone marrow cells, total bone marrow cells were similar although numbers of HSCs were modestly reduced in Ebf3-CreERT2;CXCL12^{f/-} mice injected tamoxifen once as shown in **Fig. 1c**.

e. As requested, we showed the frequencies of the populations in the bone marrow of mice and their blood counts in **Supplementary Fig. 7 and 8**.

3) *I believe that additional data is required to interpret that CXCL12 from BM LepR+/CAR cells is essential for HSC maintenance based on the results of Fig.2. 1st, LepR+ cells are also present in other organs (spleen, PMID: 26570997, etc) where they produce CXCL12. It is unclear how much locally-produced vs. plasma levels of CXCL12 are affected in young/old mice to be able to interpret the results and support the authors' conclusion. 2nd, aged mice have massively reduced endosteal niches (PMID: 31303548; PMID: 31685996) but this does not seem to be accounted for and could affect the overall degree of recombination in LepR+ cells and also explains the larger CXCL12 depletion, compared with young mice. The larger CXCL12 depletion, the less HSCs, as elegantly demonstrated by this group in the original Nature paper; but this does not demonstrate that these LepR+/CAR cells are essential to maintain HSCs through CXCL12 in any way in the young mouse. Furthermore, which cells are the authors referring to? Exclusively perisinusoidal cells as initially claimed (REF 7)? Or periarteriolar cells also marked by LepR by the same group in a later publication (PMID: 33627868)? MSCs or other bone-forming cells derived from MSCs, such as preosteoblasts, osteoblasts, osteocytes ? – all of them are targeted by LepR-Cre and therefore it is difficult to know which cell is doing what or extrapolate conclusions from overall deletion in all these different cell types producing CXCL12.*

We thank the reviewer for pointing out to an important issue.

a. We do agree that it is unclear how much locally-produced vs. plasma levels of CXCL12 are affected in young/old LepR-Cre;CXCL12^{fl/-} mice. Given that bone marrow is one of the largest organs and the expression levels of CXCL12 in CAR cells in the marrow are much higher than any other type of cells in the body, it is likely that CAR

cells are a major source of systemic plasma CXCL12 levels. Together with the facts that the numbers of CXCL12-intact CAR cells are different between young and aged LepR-Cre;CXCL12^{f/-} mice, our results, including the fact that HSC numbers were severely reduced in young tamoxifen-treated Ebf3-CreERT2;CXCL12^{f/-} mice strongly suggest that CAR cell-derived CXCL12 is essential for the maintenance of HSCs. Further studies will be needed to determine the role of LepR and CXCL12-expressing cells outside the bone marrow in HSC maintenance in young and aged animals. We added statements to the text to discuss the possibility pointed out (**Page 13, lines 290 to 295**).

b. As pointed out, there is the possibility that HSC numbers were severely reduced in aged LepR-Cre;CXCL12^{f/-} mice since aged mice have massively reduced endosteal niches. We showed that most residual HSCs were located near CXCL12-intact CAR cells but not near the bone surface in young LepR-Cre;CXCL12^{f/-} mice (**Page 7, lines 151 to 153**). Given that the expression levels of CXCL12 in CAR cells are much higher than osteoblasts, this result support the idea that most HSCs were maintained by CXCL12-intact CAR cells but not by endosteal niches in the young mutants. We added statements to the text to discuss the possibility pointed out (**Page 13, lines 290 to 295**).

c. We defined CAR cells as a fibroblast lineage that expressed higher levels of PDGFRb, CXCL12, Foxc1, and Ebf3 than any other types of cells in the body. Of note, some CAR cells do not express LepR and thus CXCL12 was deleted from less than 80% of CAR cells and HSC numbers were unaltered in young LepR-Cre;CXCL12^{f/-} mice (Ding and Morrison, Nature 495; 231, 2013). To address this issue and examine the role of CXCL12 produced by CAR cells, we analyzed Ebf3-CreERT2;CXCL12^{f/-} mice at 10 to

14 weeks after tamoxifen treatment since *Ebf3* is specifically expressed in CAR cells in the marrow (Page 5, lines 85 to 88; Seike et al., G&D 32; 359, 2018).

-[Periarteriolar *LepR*⁺ cells]

Periarteriolar cells also marked by *LepR* shown in a recent publication (PMID: 33627868) are a subset of CAR cells.

- [MSCs other than CAR cells]

MSCs other than CAR cells have been shown to be *PDGFR* α ⁺*Sca-1*⁺ cells, which surround arteries and express IL-33 using IL-33-GFP mice (Helbling et al., Cell Rep 29; 3313, 2019). These cells do not express *LepR* (Helbling et al., Cell Rep 29; 3313, 2019) or *Ebf3* (Helbling et al., Cell Rep 29; 3313, 2019; Seike et al., G&D 32; 359, 2018) and their numbers are much smaller than CXCL12-intact CAR cells in young *LepR-Cre;CXCL12*^{f/f} mice. In addition, CXCL12 mRNA levels in *PDGFR* α ⁺*Sca-1*⁺ cells are much lower than those of CAR cells (Helbling et al., Cell Rep 29; 3313, 2019).

-[preosteoblasts, osteoblasts, osteocytes]

Bone-forming cells, such as preosteoblasts, osteoblasts, and osteocytes are distinct from CAR cells but derived from CAR cells (Seike et al., G&D 32;359, 2018). Thus, as the reviewer pointed out, these cells are targeted by *LepR-Cre* in aged *LepR-Cre;CXCL12*^{f/f} mice. However, few HSCs are located near the bone surface (**Page 7, lines 151 to 153**) (Acar et al., Nature 526; 126, 2015; Upadhaya et al., Cell Stem Cell 27; 1, 2020). In addition, CXCL12 mRNA levels in osteoblasts and osteocytes are much lower than those of CAR cells (Sugiyama et al., Immunity 25; 977, 2006; Omatsu et al., Immunity 33; 387, 2010) and the numbers of osteoblast are much smaller than those of CXCL12-intact CAR cells in young *LepR-Cre;CXCL12*^{f/f} mice. These facts suggest that contribution of CXCL12 from these bone-forming cells and *PDGFR* α ⁺*Sca-1*⁺ MSCs to HSC maintenance was lower than that of CXCL12-intact CAR cells in young

LepR-Cre;CXCL12^{f/-} mice. We added statements to the text to discuss the possibility pointed out (**Page 3, lines 47 to 50; Page 7, lines 151 to 153; Page 13, lines 290 to 295**).

4) To compare the reconstitution potential of myeloid-biased or balanced (“=lymphoid-biased”) HSCs the authors should have sorted each HSC type based on the markers available and compare their function. It is very difficult to infer which HSC population has contributed to hematopoietic reconstitution after transplanting the unseparated, bulk HSCs. The frequency of CD150^{low} or CD229^{high} cells is less reduced than the frequency of overall LT-HSCs in Fig.4, so the claim that lymphoid-bias HSCs are especially affected does not seem supported. Multiple myeloid lineages were profoundly depleted as well, which does not support the overall conclusion.

As pointed out, specific markers for balanced (“=lymphoid-biased”) HSCs, by which myeloid-biased or lymphoid-biased HSCs can be separated, have not been identified. However, long-term transplantation of identical number of cells in the HSC population we performed is the completely unambiguous, stringent and reliable measure of myeloid-biased or balanced (“=lymphoid-biased”) HSC activity. The study demonstrated clearly that not only total numbers of HSCs but also the ability of HSCs to generate B cell progenitors relative to myeloid progenitors was markedly reduced in mice whose CXCL12 was deleted from all CAR cells compared with control animals (**Fig. 4**). Thus, we drew the conclusion that CAR cell-derived CXCL12 plays a specific and essential role for the maintenance of lymphoid-biased (=balanced) HSCs relative to myeloid-biased HSCs in addition to an essential role for the maintenance of the total

numbers of functional HSCs.

5) Fig4- Do they have the chimerism data throughout the transplant- Wk 4/8/12 to see if chimerism was always low or if it just dipped at the end? They also only show select subsets of cells, is there a reason why- e.g. why Pre B over other B cell populations, could they include an overall summary in the supplementary? To support ratio data, are the frequency of the populations altered in donor cells to confirm differentiation bias. Figure 4 lacks a depth of how the naive mature B cells look like in the secondary lymphoid organs such as spleen and lymph nodes. And not just reduced numbers, do they have a defect on function (i.e., antibody production after challenging with antigen, or response to LPS or anti-CD40).

a. We added the chimerism data in the peripheral blood throughout the transplant-4, 8, and 12 weeks after transplantation and the percentages of donor-derived pro-B cells, immature B cells, and mature B cells in the bone marrow at 16 weeks after transplantation. Chimerism in the peripheral blood was always low 4, 8, 12, and 16 weeks after transplantation. We stated these in the text (**Page 10, line 223 to 225**) and **Supplementary Fig. 10b**, including overall summary.

b. As requested, we showed the frequencies of hematopoietic cell populations within donor-derived cells in the bone marrow and peripheral blood of the control and mutant chimeras in **Supplementary Fig. 10c**. These support ratio data.

c. We do agree that it will be important ultimately to reveal the percentages and function of the donor-derived naive mature B cells. However, this paper focus on the role of

CAR cell-derived CXCL12 on HSCs to generate B cell progenitors. Thus, we did not examine the percentages and functions of donor-derived naive mature B cells in the secondary lymphoid organs such as spleen. However, we examined those in donor mice (tamoxifen-treated Ebf3-CreERT2;CXCL12^{f/-} mice, in which CXCL12 is deleted from all CAR cells). In contrast with severe reduction in B cell precursors in the bone marrow, the numbers of mature B cells were modestly reduced in the mutant spleen. An explanation for this discrepancy is that the mature B cells might be long-lived B cells generated before CXCL12 was deleted from bone marrow CAR cells and/or expansion of remaining B cells was increased due to reduced newly generated B cells which enter the spleen from the bone marrow. We showed the data in **Fig. 4 for review purposes.**

We believe that the generation and function of naive mature B cells in the secondary lymphoid organs in the absence of CAR cell-derived CXCL12 are beyond the scope of the present study but provide a basis for future studies.

6) Fig5- Following the culture- was there a difference in cell number at the end of the culture- how many cells were transplanted- was apoptosis rate different with/without CXCL12? Did the authors carry out FACS analysis at the end of the culture to see what cells were being transplanted? To investigate if CXCL12 affected cells differentiation and how many HSCs and other progenitors were transplanted? Was there any more FACS analysis carried out at the end of the 16-week transplant? Blood counts? Frequency of subsets?

There was no significant difference in the total cell numbers and apoptosis rate with and without CXCL12. We transplanted 75% of cells (from 82000 to 181000 cells) in a well, most of which were lin⁻c-kit⁺ primitive cells as shown in **Fig. 5 for review purposes.**

Percentages of donor-derived hematopoietic cells were variable between recipients but the donor B cell progenitor/myeloid progenitor reconstitution ratios in recipients of cultured cells with CXCL12 was significantly higher than those in recipients of cultured cells without CXCL12. We show the results of FACS analysis carried out at the end of the 16-week transplant, including blood counts and frequency of subsets in Fig. 5b and **Fig. 5 for review purposes.**

Minor comments:

- *L.70. Delete “the former mutants”*

We thanks the reviewer for the helpful comment. We deleted the words.

- *Lines 69-71: Same mouse model stated ‘Ebf3-CreERT2;CXCL12ff mice’, should one be f/-?*

We thanks the reviewer for the helpful comment. We corrected the words (Page 4, lines 65, 71, and 75).

- *Figure axis throughout could be clearer/ more detailed e.g ‘Cxcl12 mRNA (fold)’ or ‘Frequency of LT-HSCs’.*

We changed Figure axis to reflect the concerns of the reviewer.

- *Supplementary Fig2c- Could the authors show the negative GFP mouse or unstained control for the gating?*

As requested, we show the negative GFP mouse or unstained control for the gating in **Supplementary Fig. 2c.**

- *Line 121- Please briefly define ‘random spots near intact CAR cells’ in text or methods, methods reference a previous paper.*

As requested, we briefly define ‘random spots near intact CAR cells’ in methods (**Page 29, lines 647 to 649**).

- *Fig1f- Is frequency also unaltered or total cell number for context?*

Frequencies of HSCs were also unaltered and we stated this in the text (**Page 7, lines 138 and 139**) and in **Supplementary Fig. 5**.

- *When there are numerous graphs per letter in the figure, should be broken up- e.g Fig1b, Fig 2b etc*

As requested, we broke up numerous graphs per letter in the figures.

Figure 1 for review purposes

CXCL12-tdTomato^{f/f}

Figure 2 for review purposes

Figure 3 for review purposes

CAR cell

Endothelial cell

CAR cell

Ki67⁺HSC

Figure 4 for review purposes

Spleen

Figure 5 for review purposes

CXCL12(-)

CXCL12(+)

REVIEWER COMMENTS

Reviewer #1 (Remarks to the Author):

The authors have added new data that addressed most of my concerns. I particularly liked the increased data on HSC localization relative to Cxcl12 deficient versus undeleted stromal cells. These data strongly argue that the localization of HSCs is dependent upon locally produced Cxcl12 in the bone marrow, consistent with many prior studies.

My only lingering concern is that I still think the manuscript goes too far in suggesting that Cxcl12 produced by CAR/LepR+ cells preferentially regulates lymphoid-biased HSCs. As all of the reviewers noted during the first round of review, the deletion of Cxcl12 from these cells had broad effects on HSCs, myeloid, and lymphoid progenitors. Moreover, it isn't clear whether the stronger phenotype in the lymphoid lineage reflects a direct effect of Cxcl12 on these cells versus an indirect effect of depleted HSC self-renewal potential after increased cell division. The authors added a sentence to the manuscript acknowledging this possibility; however, the title and abstract didn't change much. The major conclusion remains that "Niche-derived Cxcl12 preferentially regulates lymphoid-biased HSCs". In my view, this is an unnecessary over-interpretation because the results are interesting enough, and high enough quality for publication in Nature Communications even without this more aggressive conclusion.

One other small thing for the authors to consider: they suggested in the response to reviewer comments that only 80% of CAR cells are LepR+ and that's why they had to wait so long to see a strong phenotype with Lepr-cre. Our data suggest that all CAR cells are LepR+ but that Lepr-cre doesn't recombine efficiently in LepR+ bone marrow stromal cells during the first few months after birth (see Kara et al. Developmental Cell, 2023). Consequently, high recombination efficiencies are not observed until around 3 months after birth.

Reviewer #2 (Remarks to the Author):

In the revision of the article by Nakatany et al, entitled "Niche derived Cxcl12 preferentially regulates lymphoid-biased HSCs, the authors added new transplantation data, the description of the Evi1-GFP reporter and their localization in the bone marrow niche. However, the main conclusion that Cxcl12 preferentially regulates lymphoid-biased HSCs, is still not sufficiently supported by the data. Cxcl12 deficiency results in a general defect in HSC reconstitution that affect most lineages including the lymphoid and the myeloid, both in primary and secondary transplantation. The data is more compatible with bone marrow failure given the extremely low overall level of reconstitution observed. The little

remaining reconstitution observed is likely because the mice were transplanted with a very high number of HSCs. Lineage-biased patterns of reconstitution need to be evaluated by transplanting very limiting numbers of cells (instead of the 200 HSCs that were transplanted in this study). Other points below.

1. Figure 5. The authors opted for only adding 1 example of FACS plots, instead of showing graphs summarizing the data from all mice analysed. This is because there was too much variation in the reconstitution. How representative are the examples shown then? To describe a bias in reconstitution the authors should show % of donor cell contribution within each of the populations, instead of a frequency of each population in total donor. Graphs with summary data from all mice and statistical analysis is needed.
2. Total LSK SALM population is not different (when excluding CD34 as a marker) suggesting that result might indeed be due to CD34 up-regulation. In light of this, the authors should have used other markers like EPCR, that are more stable to support their conclusions.
3. Supplementary figure 2-3 – It still remains unclear if the Evi1-GFP reporter allows the analysis of HSCs with the purity described. The authors fixed cells on a slide to propose that fixation kills the GFP in MPPs but not in HSCs. It is unclear if this replicates what happens in tissue sections. Sup Fig.2d shows that only a very small minority of the total Evi1-GFP+Kit+ cells are indeed HSCs and the threshold used to gate the Evi1-GFP^{Hi} cells is largely arbitrary. The authors state in their rebuttal that in their immunofluorescence analysis of the bone marrow, they analysed 960 HSCs from controls and 1030 HSCs from mutants? Based on the HSC purity the authors report here, hundreds of mice must have been analysed. How many HSCs per section were observed and how many mice were analysed then?

Minor:

- Fig4e – What is the meaning of the PreB/HSC and ProB/HSC ratios?
- Fig 4.g - Show representative gating for these populations. There is a difference of 5%. Is this meaningful?
- Sup Fig 10 is confusing. Better explanation of the graph axes is needed. What is the difference between panel b and Fig4a-b
- Better referencing of the figures is needed. It is frequently left to the reviewer to guess which panel/graphs within the figure the authors are referring to.
- Line 224 states that PB analysis was done at 4,8,12 and 16 weeks was done but only 1 time point is shown.

Reviewer #3 (Remarks to the Author):

The authors have sufficiently addressed the comments. Although the use of available markers to discriminate myeloid-biased (e.g. CD150^{hi}, Wvf) and balanced HSCs would have provided more clear evidence of direct and indirect effects, I understand that these are time-consuming experiments that should not detract from reporting these interesting findings in a timely fashion.

Reviewer #1:

We thank the reviewer for reviewing this manuscript and for the helpful comments.

The authors have added new data that addressed most of my concerns. I particularly liked the increased data on HSC localization relative to Cxcl12 deficient versus undeleted stromal cells. These data strongly argue that the localization of HSCs is dependent upon locally produced Cxcl12 in the bone marrow, consistent with many prior studies.

My only lingering concern is that I still think the manuscript goes too far in suggesting that Cxcl12 produced by CAR/LepR+ cells preferentially regulates lymphoid-biased HSCs. As all of the reviewers noted during the first round of review, the deletion of Cxcl12 from these cells had broad effects on HSCs, myeloid, and lymphoid progenitors. Moreover, it isn't clear whether the stronger phenotype in the lymphoid lineage reflects a direct effect of Cxcl12 on these cells versus an indirect effect of depleted HSC self-renewal potential after increased cell division. The authors added a sentence to the manuscript acknowledging this possibility; however, the title and abstract didn't change much. The major conclusion remains that "Niche-derived Cxcl12 preferentially regulates lymphoid-biased HSCs". In my view, this is an unnecessary over-interpretation because the results are interesting enough, and high enough quality for publication in Nature Communications even without this more aggressive conclusion.

We thank the reviewer for the positive comments.

As requested, we added the suggested sentence to the text (**Page 14, lines 323 to 325**) and changed the title, abstract, and text more significantly (**Page 1; page 2, lines 25, 26, and 31 to 33; Page 4, lines 84 to 85; Fig. 5d, and the legend for Fig. 5d**).

One other small thing for the authors to consider: they suggested in the response to reviewer comments that only 80% of CAR cells are LepR⁺ and that's why they had to wait so long to see a strong phenotype with Lepr-cre. Our data suggest that all CAR cells are LepR⁺ but that Lepr-cre doesn't recombine efficiently in LepR⁺ bone marrow stromal cells during the first few months after birth (see Kara et al. Developmental Cell, 2023). Consequently, high recombination efficiencies are not observed until around 3 months after birth.

We agree. We stated this in the text (**Page 4, Lines 72 to 74**).

Reviewer #2:

We thank the reviewer for reviewing this manuscript and for the helpful comments.

In the revision of the article by Nakatany et al, entitled “Niche derived Cxcl12 preferentially regulates lymphoid-biased HSCs, the authors added new transplantation data, the description of the Evi1-GFP reporter and their localization in the bone marrow niche. However, the main conclusion that Cxcl12 preferentially regulates lymphoid-biased HSCs, is still not sufficiently supported by the data. Cxcl12 deficiency results in a general defect in HSC reconstitution that affect most lineages including the lymphoid and the myeloid, both in primary and secondary transplantation. The data is more compatible with bone marrow failure given the extremely low overall level of reconstitution observed. The little remaining reconstitution observed is likely because the mice were transplanted with a very high number of HSCs. Lineage-biased patterns of reconstitution need to be evaluated by transplanting very limiting numbers of cells (instead of the 200 HSCs that were transplanted in this study). Other points below.

We agree with the reviewer that the numbers of HSCs were markedly reduced in CXCL12^{ACAR} mice, in which CXCL12 were deleted from most CAR cells. However, the results of the experiments using primary and secondary transplantation and the limiting-dilution competitive repopulating units (CRUs) assay clearly demonstrated that significant numbers of functional long-term HSCs were present in CXCL12^{ACAR} mice (**Fig. 3f and Supplementary Fig. 9 and 12**).

We believe that results from the transplantation of 200 HSCs showed averages of ability of HSCs to produce B cell progenitors relative to myeloid progenitors and demonstrated that they were significantly reduced in CXCL12^{ACAR} mice.

Transplantation of very limiting numbers of cells will show the frequencies of lymphoid-biased HSCs and myeloid-biased HSCs in the HSC population; however, our claim is that averages of ability of HSCs to produce B cell progenitors relative to myeloid progenitors were reduced in CXCL12^{ACAR} mice. We anticipate taking more than 4 months and we believe that importance of our findings will not be changed without these new experiments, providing a basis for future mechanistic studies.

1. Figure 5. The authors opted for only adding 1 example of FACS plots, instead of showing graphs summarizing the data from all mice analysed. This is because there was too much variation in the reconstitution. How representative are the examples shown then? To describe a bias in reconstitution the authors should show % of donor cell contribution within each of the populations, instead of a frequency of each population in total donor. Graphs with summary data from all mice and statistical analysis is needed.

As requested, we show graphs summarizing % of donor cell contribution within each of the populations (**Supplementary Fig. 13**). Since the numbers of functional HSCs were variable between wells after culture, percentages of donor-derived hematopoietic cells were variable between recipients transplanted with cultured HSCs; however, the donor B cell progenitor/myeloid progenitor reconstitution ratios in recipients of cultured cells with CXCL12 was significantly higher than those in recipients of cultured cells without CXCL12 (**Fig. 5c**). In addition, we show another example of FACS plots of another recipient in **Fig. 1 for review purposes**.

2. Total LSK SALM population is not different (when excluding CD34 as a marker) suggesting that result might indeed be due to CD34 up-regulation. In light of this, the

authors should have used other markers like EPCR, that are more stable to support their conclusions.

We thank the reviewer for the helpful comments. We added the data showing the frequencies and numbers of EPCR⁺CD150⁺CD48⁻LSK (LSK-ESLAM) HSCs in tamoxifen-treated Ebf3-CreERT2;CXCL12^{f/-} mice, in which CXCL12 was deleted from more than 99.5% of the CAR cells (CXCL12^{ΔCAR} mice) (**Fig. 3c and Supplementary Fig. 8a**). LSK-ESLAM HSCs were reduced but present in CXCL12^{ΔCAR} mice. In addition, we replaced the data using CD34 with the data analyzing the LSK-SLAM population in **Fig. 4e**.

3. Supplementary figure 2-3 – It still remains unclear if the Evi1-GFP reporter allows the analysis of HSCs with the purity described. The authors fixed cells on a slide to propose that fixation kills the GFP in MPPs but not in HSCs. It is unclear if this replicates what happens in tissue sections. Sup Fig.2d shows that only a very small minority of the total Evi1-GFP+Kit+ cells are indeed HSCs and the threshold used to gate the Evi1-GFP^{hi} cells is largely arbitrary. The authors state in their rebuttal that in their immunofluorescence analysis of the bone marrow, they analysed 960 HSCs from controls and 1030 HSCs from mutants? Based on the HSC purity the authors report here, hundreds of mice must have been analysed. How many HSCs per section were observed and how many mice were analysed then?

We agree with the thoughts of the reviewer concerning fixed Evi1-GFP⁺c-kit⁺ cells; however, we believe that the most reliable method to determine if Evi1-GFP⁺c-kit⁺ cells are HSCs or MPPs on bone marrow sections is to analyze the GFP expression of sorted

HSCs and MPPs that were fixed on a slide. Since these experiments showed that the fluorescence signals were detected in HSCs but not in MPPs, Evi1-GFP⁺c-kit⁺ cells detected in the fixed bone marrow sections would be included in the HSC population.

Flow cytometric analysis show that the average number of Evi1-GFP⁺c-kit⁺ cells is approximately 5800 in two femurs and tibiae at 21-25 weeks of age and we analyzed 50 bone marrow sections (15-25 Evi1-GFP⁺c-kit⁺ cells per section) from the femurs of four control mice and 52 bone marrow sections (15-25 Evi1-GFP⁺c-kit⁺ cells per section) from the femurs of four mutant mice. We stated these in the **legend for Fig. 1h-l (Page 17, line 398)**. The frequencies of Evi1-GFP⁺c-kit⁺ cells in c-kit⁺ cells in bone marrow sections (approximately 0.116 %) were similar to the frequencies of Evi1-GFP^{hi}c-kit⁺ cells in c-kit⁺ cells shown by flow cytometry (approximately 0.154%). These results support the idea that Evi1-GFP⁺c-kit⁺ cells were phenotypic HSCs in bone marrow sections.

Minor:

- *Fig4e – What is the meaning of the PreB/HSC and ProB/HSC ratios?*

Pre-B/HSC and Pro-B/HSC mean % donor Pro-B cells or Pre-B cells divided by % donor LSK-SLAM HSCs (CD150⁺CD48⁻ subset of LSK), respectively. Thus, we stated this (**Page 11, lines 240 to 242**) and replaced the words Pre-B/HSC and Pro-B/HSC with Pre-B/LSK-SLAM and Pro-B/LSK-SLAM, respectively (**Fig. 4e; the legend for Fig. 4e**).

- *Fig 4.g - Show representative gating for these populations. There is a difference of 5%. Is this meaningful?*

As requested, we show representative gating for the CD150^{lo} or CD229^{hi} subset of LSK-SLAM cells (**Supplementary Fig. 11**). Specific markers for lymphoid-biased (= balanced) HSCs, by which lymphoid-biased or myeloid-biased HSCs can be separated, have not been identified; however, frequencies of lymphoid-biased HSCs were larger in CD150^{lo} or CD229^{hi} subset than those in the other subsets within LSK-SLAM cells (ref. 23 and 29). Therefore, we think that significant reductions in frequencies of CD150^{lo} or CD229^{hi} subset in LSK-SLAM cells albeit to a lesser extent are consistent with our conclusions and thus would be meaningful.

- Sup Fig 10 is confusing. Better explanation of the graph axes is needed. What is the difference between panel b and Fig4a-b.

We thank the reviewer for the helpful comment. While addressing this comment, we realized an error in the graph axis. We have corrected this error (**Supplementary Fig. 10b**). Sup Fig 10b is an overall summary of frequencies of donor-derived cells in various hematopoietic cell populations in the recipients of HSCs from CXCL12^{ACAR} mice, including the data shown in Fig. 4a and 4b.

- Better referencing of the figures is needed. It is frequently left to the reviewer to guess which panel/graphs within the figure the authors are referring to.

We agree with the reviewer's comment and stated these in the text (**Page 10, lines 229, 230, 232, and 234**).

- Line 224 states that PB analysis was done at 4,8,12 and 16 weeks was done but only 1 time point is shown.

We show PB analysis done at 4,8,12 and 16 weeks in **Supplementary Fig. 9**.

Reviewer #3:

The authors have sufficiently addressed the comments. Although the use of available markers to discriminate myeloid-biased (e.g. CD150^{hi}, W^vf) and balanced HSCs would have provided more clear evidence of direct and indirect effects, I understand that these are time-consuming experiments that should not detract from reporting these interesting findings in a timely fashion.

We thank the reviewer for reviewing this manuscript and for the positive comments.

Figure 1 for review purposes

Donor-derived cells

REVIEWERS' COMMENTS

Reviewer #1 (Remarks to the Author):

The authors have done a good job of revising the manuscript to address the concerns about lymphoid biased HSCs. I have only two remaining suggestions, both of which are easily addressed.

1. I would suggest eliminating the word "fibroblastic" from the title and abstract. This new word was introduced in the last draft for the first time and could be deleted without changing the meaning of either the title or the abstract. The reason I believe it should be deleted is that there is enormous confusion in the HSC niche field every time a new word is used to describe the niche. The reality is that there is a single type of niche, associated with sinusoidal blood vessels, that depends upon growth factors from CAR/LepR+ cells and endothelial cells. However, when different words are used to describe this niche, confusion is created about whether the authors are describing the same niche with a different name or a different niche. Some people will believe the "fibroblastic" niche is different from the perivascular niche that the Nagasawa lab and other labs have described in many prior studies. In my view, there is no reason to introduce another new term to describe the same cells/same niche. The manuscript does not address the question of whether CAR/LepR+ cells are fibroblasts or not and I'm not aware of any prior study that has addressed this question so it's not clear to me why the authors introduced this new claim into the title.

2. The key data in the paper were obtained by using Ebf3-creER and Lepr-cre to delete Cxcl12 from BM stromal cells but this central information is currently missing from the abstract. This could be fixed by changing a single word in line 27 ("CAR" to "Ebf3+/LepR+") of the abstract so that sentence reads "In mice whose Cxcl12 is deleted from all Ebf3+/LepR+ cells, HSCs were markedly reduced..."

Reviewer #2 (Remarks to the Author):

The authors have added new data that significantly improved the manuscript. While the involvement of Cxcl12 in HSC regulation has been now well explored and brings new important clarifications I feel that the specific role of Cxcl12 in regulating lymphoid-biased HSCs is not definitive, based on very minor differences and on data that has very high variability (Ext.Data Fig13). Previous studies (PMID: 29298288) have shown that in transplantation and at single cell level, lymphoid potential does not exist separately from myeloid and MK/E potentials in HSCs. This challenges the existence of lymphoid-biased HSCs and the authors should discuss their findings in light of this study. Some other minor points below.

- Define genotypes of control mice throughout the manuscript. Are the control mice Cre+ and tamoxifen treated?
- Fig1i – Indicate in figure legend the numbers of cells analysed in each condition.
- Fig4c – Should this be “Chimeric BM”?
- Fig4e – The authors still need to clarify what is the biological meaning of a PreB/LT-HSC or ProB/LT-HSC ratios.
- Fig4h should also include the GMP/MEP ratios as donor contribution towards GMP and MEP also seems to be different (Fig4a).
- Fig5b should show CD45.1 vs CD45.2 (or frequency of donor-derived cells) for each population rather than relative frequencies of different populations, as the last don't access lineage biases.
- Fig5c should also include the Pro-B/GMP ratio, which is used throughout the manuscript in addition to the Pre-B/GMP ratio.

Reviewer #1:

The authors have done a good job of revising the manuscript to address the concerns about lymphoid biased HSCs. I have only two remaining suggestions, both of which are easily addressed.

1. I would suggest eliminating the word "fibroblastic" from the title and abstract. This new word was introduced in the last draft for the first time and could be deleted without changing the meaning of either the title or the abstract. The reason I believe it should be deleted is that there is enormous confusion in the HSC niche field every time a new word is used to describe the niche. The reality is that there is a single type of niche, associated with sinusoidal blood vessels, that depends upon growth factors from CAR/LepR⁺ cells and endothelial cells. However, when different words are used to describe this niche, confusion is created about whether the authors are describing the same niche with a different name or a different niche. Some people will believe the "fibroblastic" niche is different from the perivascular niche that the Nagasawa lab and other labs have described in many prior studies. In my view, there is no reason to introduce another new term to describe the same cells/same niche. The manuscript does not address the question of whether CAR/LepR⁺ cells are fibroblasts or not and I'm not aware of any prior study that has addressed this question so it's not clear to me why the authors introduced this new claim into the title.

We thank the reviewer for the positive and helpful comments.

We agree. As requested, we eliminated the word "fibroblastic" from the title and abstract (**Page 1, line 1; Page 2, line 25**).

2. The key data in the paper were obtained by using Ebf3-creER and Lepr-cre to delete Cxcl12 from BM stromal cells but this central information is currently missing from the abstract. This could be fixed by changing a single word in line 27 ("CAR" to "Ebf3+/LepR+") of the abstract so that sentence reads "In mice whose Cxcl12 is deleted from all Ebf3+/LepR+ cells, HSCs were markedly reduced..."

We agree and added the word “Ebf3⁺/ leptin receptor (LepR)⁺” in the abstract (**Page 2, line 27**).

Reviewer #2 (Remarks to the Author):

The authors have added new data that significantly improved the manuscript. While the involvement of Cxcl12 in HSC regulation has been now well explored and brings new important clarifications I feel that the specific the role of Cxcl12 in regulating lymphoid-biased HSCs is not definitive, based on very minor differences and on data that has very high variability (Ext.Data Fig13). Previous studies (PMID: 29298288) have shown that in transplantation and at single cell level, lymphoid potential does not exists separately from myeloid and MK/E potentials in HSCs. This challenges the existence of lymphoid-biased HSCs and the authors should discuss their findings in light of this study. Some other minor points below.

We thank the reviewer for the positive and helpful comments. As requested, we discuss the findings shown by previous studies challenging the existence of lymphoid-biased HSCs (**Page 14, lines 325 to 329 and 330**) and added the word “balanced HSCs” in the abstract (**Page 2, line 34**).

- Define genotypes of control mice throughout the manuscript. Are the control mice Cre+ and tamoxifen treated?

As requested, we defined genotypes of control mice throughout the manuscript (**Page 26, line 594; Page 26, line 601; Page 27, lines 617, 626, 632, and 639; Page 28, lines 646, 648, 652, 659, and 665; Page 29, lines 667, 675, 678, 680, 681, and 686 in the manuscript**). (**In the supplementary information, Page 5, line 4; Page 9, lines 3 and**

5; Page 11, lines 3 and 5, Page 12, lines 3, 9, and 11; Page 14, lines 3, 7, 14, and 16; Page 16, line 4; Page 16, line 4).

- Fig1i – Indicate in figure legend the numbers of cells analysed in each condition.

As requested, we indicated the numbers of cells analyzed in each condition in figure legends (**Page 26, line 616; Page 27, lines 619 to 620**).

- Fig4c – Should this be “Chimeric BM”?

We thank the reviewer for the helpful comment. We realized two errors in Fig. 4c and 4f and have corrected them (**Page 10, lines 230 and 233; Fig. 4c and 4f**).

- Fig4e – The authors still need to clarify what is the biological meaning of a PreB/LT-HSC or ProB/LT-HSC ratios.

The results that donor PreB/LT-HSC or ProB/LT-HSC reconstitution ratios were markedly reduced in wild-type recipients transplanted with HSCs from CXCL12^{ACAR} mice suggest that frequencies of HSCs, which generated B cell progenitors were reduced and/or the ability of HSCs to generate B cell progenitors was reduced to a larger extent than the ability of HSCs to self-renew in CXCL12^{ACAR} mice compared with wild-type mice. We stated these in the text (**Page 14, lines 333 to 338**)

- Fig4h should also include the GMP/MEP ratios as donor contribution towards GMP and MEP also seems to be different (Fig4a).

As requested, we show the GMP/MEP ratios in **Fig. 4h**. Donor contribution towards GMP and MEP also seems to be similar.

- Fig5b should show CD45.1 vs CD45.2 (or frequency of donor-derived cells) for each population rather than relative frequencies of different populations, as the last don't access lineage biases.

As requested, we show CD45.1 vs CD45.2 (or frequency of donor-derived cells) for each population in **Fig. 5b**.

- Fig5c should also include the Pro-B/GMP ratio, which is used throughout the manuscript in addition to the Pre-B/GMP ratio.

As requested, we show the Pro-B/GMP ratio in Fig. 5 (**Fig. 5d**).